# Encoding performance of cortical neurons critically depends on their morphological and neurophysiological properties

Omer Revah[1]*, Fred Wolf[2,3,4,5,6,7], Michael J. Gutnick[1]*, Andreas Neef[2,3,4,5,6,7]*

1 Koret School of Veterinary Medicine, Robert H. Smith Faculty of Agriculture, Food, and Environment, The Hebrew University of Jerusalem, Rehovot, Israel, 2 Göttingen Campus Institute for Dynamics of Biological Networks, Göttingen, Germany, 3 Max Planck Institute for Dynamics and Self-Organization, Göttingen, Germany, 4 Bernstein Center for Computational Neuroscience, Göttingen, Germany, 5 Max Planck Institute for Multidisciplinary Sciences, Göttingen, Germany, 6 Institute for Nonlinear Dynamics, Georg-August-University Göttingen, Göttingen, Germany, 7 Center for Biostructural Imaging of Neurodegeneration, Göttingen, Germany

* omer.revah@mail.huji.ac.il (OR); michael.gutnick@mail.huji.ac.il (MJG); aneef@gwdg.de (AN)

## Abstract

Sixty years after the concept of population coding in neuronal networks was introduced, we still lack a comprehensive understanding of its performance limits and the role of neuronal physiology. Here, we use dynamic gain analysis in a general model of population coding and demonstrate that disparate parameters of neurons and populations determine how accurately they can encode information. These are cell number, cell size, and the correlation time of the background noise. We experimentally test and confirm these predictions on neurons of excitatory populations in the mouse barrel cortex. Surprisingly, dendrite size and background correlations are precisely matched with the number of neurons in layer 4, such that even a single thalamocortical spike at the input is reliably reflected in the population output. However, this encoding performance can be modulated by the channels that mediate M-current, suggesting that coding in layer 4 may vary as a function of brain state.

## Introduction

A major challenge of modern neuroscience is to understand how information is encoded and transferred within and between neuronal circuits given the circuit infrastructure, including the morphology, connectivity, molecular properties, and physiological characteristics of the constituent neurons. In mouse cortex, individual neurons fire no more than a few action potentials (APs) each second [1–4], so information is encoded as the combined spike output of subpopulations of neurons responding to common inputs [5,6]. The encoding capability of a population may be inferred from the dynamic gain curves of its constituent neurons. This is a measure of a neuron's ability to lock its spikes onto the frequency components of its input (Box 1, S1A and

**Data availability statement:** All data are available in the main text, the Supporting information files, or the data and code repository https://doi.org/10.25625/VQUKFU.

**Funding:** This work was supported by the Ministry for Science and Culture of Lower Saxony (MWK) (https://www.mwk.niedersachsen.de/) and VolkswagenStiftung (https://www.volkswagenstiftung.de/en) through the program "Niedersächsisches Vorab" to FW. Leibniz Association (https://www.leibniz-gemeinschaft.de/en) (project K265/2019) (to FW) Deutsche Forschungsgemeinschaft (DFG, German Research Foundation https://www.dfg.de/en) Project-ID 436260547, in relation to NeuroNex (NSF 2015276) (to FW and AN) as well as, Project-ID 430156276 (SPP 2205), Project-ID 454648639 (SFB 1528), Project-ID 273725443 – (SPP 1782), Project-ID 317475864 (CRC 1286) and through DFG under Germany's Excellence Strategy - EXC 2067/1-390729940 (to FW), Project ID 528760423 (CRC 1690) (to AN) VW Foundation (https://www.volkswagenstiftung.de/en) grant ZN2632 (FW), and GIF (https://www.gif.org.il) (906-17.1/2006) (to MJG and FW). The funders played no role in study design, data collection and analysis, nor in the preparation of the manuscript and the decision to publish.

**Competing interests:** The authors have declared that no competing interests exist.

**Abbreviations:** aCSF, artificial cerebrospinal fluid; aEPSC, artificial EPSC; AIS, axon initial segment; APs, action potentials; DC, direct current; EPSCs, excitatory post-synaptic currents; IPSCs, inhibitory post-synaptic currents; OU, Ornstein–Uhlenbeck; STA, spike-triggered average.

S1B Fig). Thus, it represents the temporal precision with which the combined firing rate of the neurons can track changes in a common input stream [6] (see S1G and S1H Fig). Numerous theoretical studies have investigated the basic principles of population coding. Early work by Knight [7] demonstrated that the information present in the common input can be perfectly encoded by an infinitely large population of noisy, instantaneously responding neurons. However, the size of cortical populations is finite and actual neurons do not respond instantaneously. As a result, the individual neuron's responses are jittered, which leads to information loss that may have dire consequences. Too much jitter, or, in the language of dynamic gain analysis, a too-narrow bandwidth of the dynamic gain, results in chaotic cortical dynamics [8]. A number of theoretical studies identified which features constrain the bandwidth, and thereby, the quality of encoding by a finite population of biophysically-inspired, non-perfect neuron models. These critical features include the correlation time of the background noise [9], the dynamics of AP generation [9,10], and the dendritic size of the neurons involved [11,12]. Dendrites amplify the impact of rapid input fluctuations because they shunt more slowly changing inputs [11,13]. Taken together, these studies predict that the encoding performance of a neuronal population reflects multiple factors, including the number of neurons involved, their morphology, the kinetic properties of the synaptic activity responsible for the background noise, and the ionic mechanisms governing AP generation. The relative importance of each of these elements can only be determined experimentally.

Experimental assessment of these theoretical concepts requires a stable population of well-defined neurons, in which each of the above factors is known. However, a cortical neuron may participate in many functional networks, and at any moment, a relevant subpopulation of neurons is sparsely embedded within a larger network (Fig 1A). Moreover, the characteristics of the common input are usually unknown. Here, we overcome these problems by studying the encoding performance of a stable population of well-described cortical neurons: the excitatory cells of Layer 4 (L4) of the mouse somatosensory cortex [14–18]. Numbers of neurons in this population, and their input and output, are known [19,20]. The common input is via projections of thalamocortical relay cells [20,21], accounting for about 15% of the excitatory synapses in spiny stellate cells [22]. The background noise reflects activities of recurrent connections among other L4 cells of the same barrel [17,21], and the population output converges onto pyramidal cells in Layers 2, 3, and 5 [17]. L4 is thus a relay layer, rapidly informing the rest of the cortical column about changes in tactile input. However, because L4 neurons have extremely short dendrites that span no more than a few hundred microns [23–25], we expect their encoding capacity to be limited [11–13]. Here, we experimentally compare the encoding bandwidth of L4 neurons to that of the much larger Layer 5 (L5) pyramidal neurons and find that, as predicted by theory, it is reduced, given the same noise parameters and population size. However, we now also demonstrate that synaptic receptor kinetics, axonal potassium channel properties, and population size combine to compensate for the morphological constraints. Indeed, all these factors appear to be matched, such that a single thalamocortical AP is reliably reflected in the L4 population output.

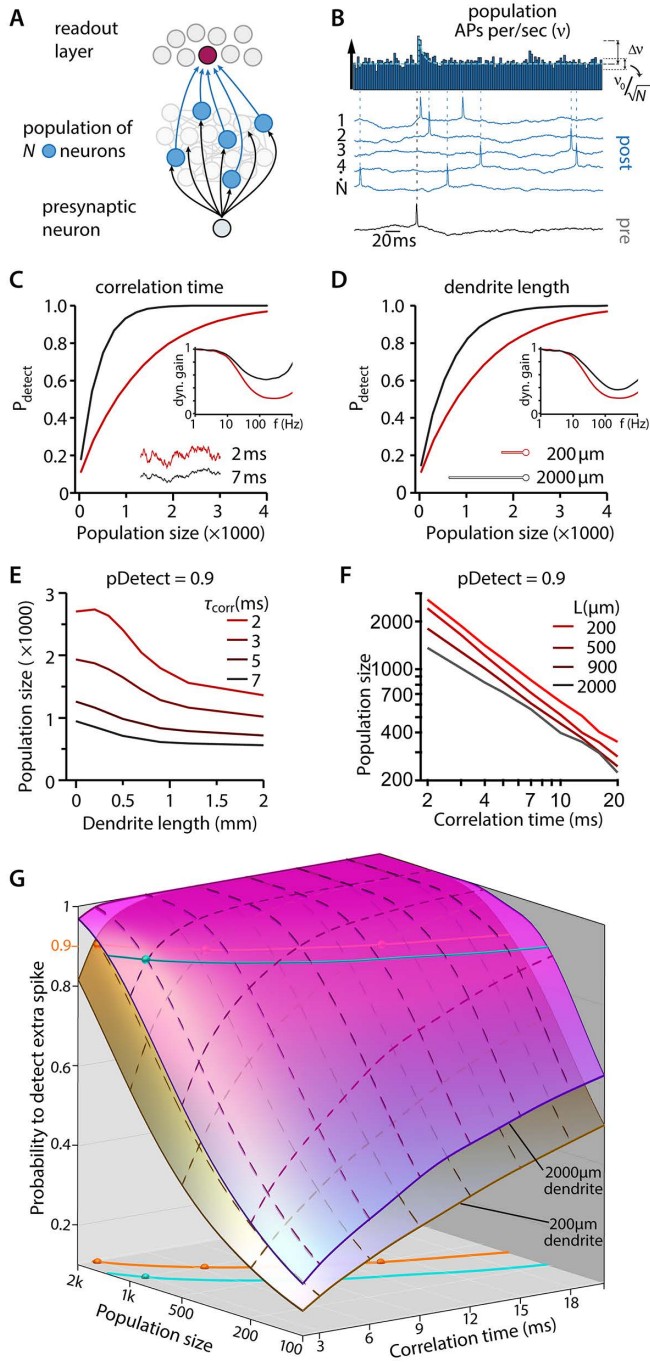

**Fig 1. Encoding and decoding in a model population of neurons. (A)** Schematics showing a simplified neural network comprised of a single pre-synaptic neuron connected to 'N' post-synaptic neurons firing asynchronously at about 3 Hz, and a readout layer which detects firing rate changes in the post-synaptic population. **(B)** Brief increases in the population firing rate ($\Delta v$) occur when a single AP-induced excitatory synaptic current is added to the total input. The probability of detection (pDetect) by an ideal decoder (Materials and methods) in the readout-layer depends on the relationship between $\Delta v$ and the basal firing rate fluctuations ($v_0/\sqrt{N}$). **(C)** The effect of background synaptic noise correlation time on pDetect and on dynamic gain. **(D)** The effect of dendrite length on pDetect and on dynamic gain. **(E, F)** pDetect reflects relationships between population size, dendrite length, and synaptic noise correlation time. **(G)** Summary of the relationship between properties of recurrent synaptic connections (correlation time), size of the encoding population N, and the size of the dendrites L. Shown are surfaces for 200 and 2,000 µm long dendrites. Combinations of the three parameters that allow for a 90% detection probability are indicated by colored lines (blue for L = 2,000 µm, orange for L = 200 µm).

## Box 1. Dynamic gain.

Through its synapses, a cortical neuron receives hundreds to thousands of inputs per second, so its membrane potential is constantly fluctuating. These fluctuations drive the cell to fire APs. The dynamic gain function captures the relationship between the different frequency components that constitute the input, e.g., slow undulations or more abrupt fluctuations, and the times at which APs are initiated. This explanation directs a method to measure dynamic gain: i) inject a random fluctuating current that drives irregular firing at a pre-decided rate, ii) superimpose a weak sinusoidal current, and iii) determine how strongly the AP generation is phase-locked to the sinusoid (see Materials and methods: Vector strength and S1A–S1C Fig). Thus, frequencies that cause stronger phase-locking are amplified in the neuron's output. In this sense, the dynamic gain function is essentially a tuning curve, since it captures how the neuronal output reflects the various brain rhythms and conveys them to downstream neurons.

The dynamic gain not only exposes the neurons to brain rhythms, it also exposes how rapidly the cell can respond with an AP. Thus, a wider bandwidth of the dynamic gain function results in an earlier, stronger, and narrower response. In our study, we used the dynamic gain function to reflect encoding by a neuronal population. When several neurons that contact the same downstream neuron, receive a common input, this population's collective encoding can be captured by averaging the individual neurons' (complex-valued) dynamic gain functions, weighted by each neuron's firing rate.

An alternative method to derive the dynamic gain function does not require an additional sinusoidal input (see Materials and methods: Dynamic gain calculation and S1D and S1E Fig) because the stochastically fluctuating input is, itself, composed of various frequency components. The results of these two approaches are not only theoretically identical [26] but also identical in our experiments (S1E Fig).

## Results

### Identification of the factors that influence coding capabilities, as predicted by theory

We created a simple model to examine the influence of cellular parameters on a neuronal population's coding capability. It captures how diverging, feedforward input is relayed through a population of neurons whose output converges onto readout neurons (Fig 1A, black arrows onto blue relay neurons and blue arrows onto red readout neurons). Each neuron consists of a soma and a dendrite of length $L$. Each neuron has two distinct sources of input: extrinsic projections and massive local, recurrent connectivity. In cortical networks, the post-synaptic effect of local, recurrent inputs reflects a balance of inhibition and excitation, and it is largely asynchronous from neuron to neuron. Because we study how the population's output reflects its common feedforward input, we refer to the recurrent, asynchronous input as background noise. We represented this input to each neuron as an independent, stochastically fluctuating current with correlation time $\tau$. Its amplitude was chosen to evoke irregular AP firing at a basal rate $v_0 = 3$ Hz (Fig 1B). As detailed in the Materials and methods, this model captures how dendrite size and synaptic kinetics impact the encoding of input by an ideal, instantaneous AP generator.

Modeled on the connectivity of L4, the relevant information enters into a subpopulation of $N$ neurons through diverging feedforward input, where it evokes synaptic currents that appear in synchrony at each neuron in the subpopulation (Fig 1A, black arrows into blue neurons). When the $N$ neurons receive synaptic currents triggered by an individual presynaptic AP, the population's firing rate momentarily increases ($\Delta v$ in Fig 1B). How salient this minimal input is for the readout neuron depends on the size and duration of $\Delta v$. We analyzed this using an "optimal decoder" (see Materials and methods),

 

which compares the rate increase during a 3 ms time window against the rate fluctuations that precede the common input. The decoder returns a detection probability (pDetect, see Materials and methods).

This analysis reproduced the dependencies expected from previous experimental and theoretical studies: larger dendrites (*L*) and longer noise correlation times (τ) improved the dynamic gain at high frequencies (insets in Fig 1C and 1D). This, in turn, increases Δ*v* (see also S1G–S1J Fig), which improves the detectability of the single afferent AP. Enlarging the population size (*N*) was also beneficial, because it reduced the fluctuations in population rate (Fig 1B, 1E, and 1F). In an actual neural population, the coding efficiency will reflect some combination of these three parameters (Fig 1G). The model predicts that for a fixed correlation time of 5 ms, populations of neurons with short (200 μm) dendrites require 1,200 cells to achieve 90% detection probability, whereas only 700 are required when dendrites are long (2,000 μm). Further, the model predicts that the encoding performance of 1,000 cells with long dendrites and 3 ms correlated input (cyan dot in Fig 1G) can be matched by cells with short dendrites by either doubling the number of cells to 2,000 or by increasing the correlation time to about 7 ms. A further increase to 14 ms would even allow the same performance by only 500 cells (3 orange dots in Fig 1G). The model thus predicts that in order to achieve high-performance encoding, neurons must either be larger, which requires more space, or have slower temporal dynamics of the background noise.

### L4 neurons are less able than L5 neurons to code high frequencies when the background correlation time is the same

We sought to understand how these theoretical concepts of population encoding are implemented by actual cortical neurons and cortical circuits. Excitatory neurons in L4 and L5 differ dramatically in size (Fig 2A). Most excitatory L4 neurons are spiny stellate cells; their dendritic arbors are short and limited to the same layer and cortical column [16,19,20,24,25,27,28]. The L5 pyramidal cells, by contrast, have broad basal dendritic skirts and large apical dendrites that extend throughout the cortical depth [29,30]. To determine how these two cell types compare electrophysiologically,

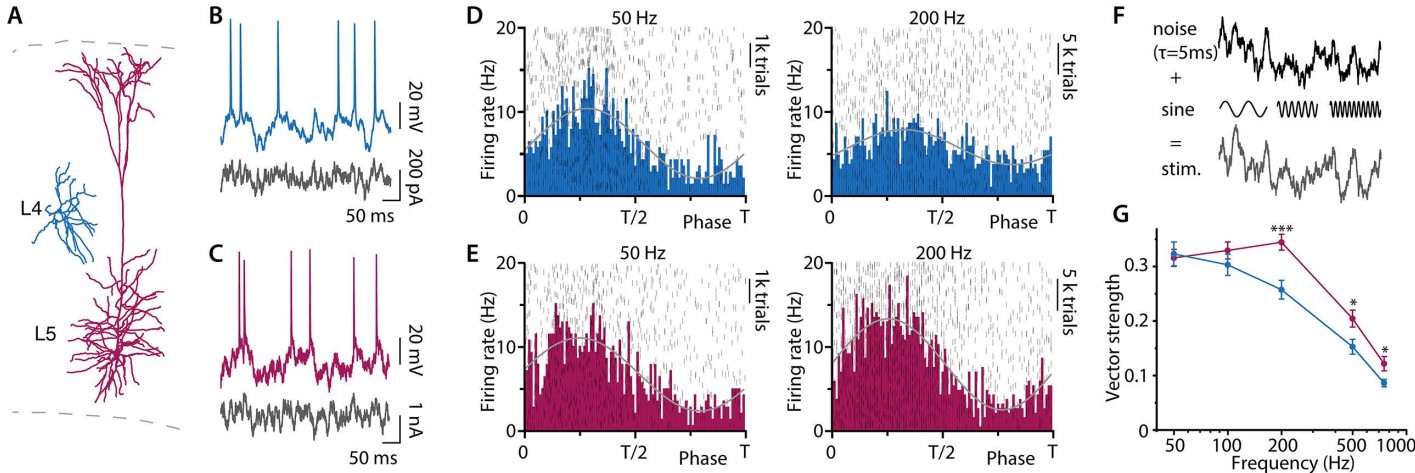

**Fig 2. When temporal characteristics of the noise are the same, L4 neurons are less effective than L5 neurons at encoding high frequencies.**
**(A)** Laminar position and morphology of L4 (blue) and L5 (red) excitatory neurons in the somatosensory cortex. **(B)** and **(C)** Voltage fluctuations and neuronal firing in response to current noise injections (black) in a L4 cell (B) and a L5 pyramidal cell (C). **(D)** and **(E)** Raster plot (gray) and histogram of AP times as a function of phase of the embedded sine wave (50 and 200 Hz) for L4 cells (D) and L5 pyramidal cells (E). Data in raster plots and histograms pooled from 3 neurons for each group. **(F)** For both cells, stimuli are the sum of a fluctuating waveform with 5-ms correlation time and a small sine wave of 50–750 Hz frequency. **(G)** Vector strength as a function of input frequency (50 Hz, *n* = 12 L4, *n* = 13 L5, *p* = 0.79; 100 Hz, *n* = 13 L4, *n* = 13 L5, *p* = 0.30; 200 Hz, *n* = 13 L4; *n* = 13 L5, *p* = 0.0008; 500 Hz, *n* = 13 L4; *n* = 13 L5, *p* = 0.020; 750 Hz, *n* = 12 L4; *n* = 13 L5, *p* = 0.026). Shown are means ± SE. *p*-values result from *t* test (see Materials and methods). Data https://doi.org/10.25625/VQUKFU.

we performed whole-cell patch clamp recordings from identified L4 and regular-firing L5 neurons in acute brain slices of the mouse somatosensory cortex. Table 1 summarizes the passive and active membrane properties of the cells. L4 neurons had approximately 2.5-fold lower membrane capacitance, 6-fold higher input resistance, and 2-fold longer membrane time constant, consistent with their compact morphology [28]. As noted above, our theoretical model predicts that the smaller L4 neurons should be less able to encode high frequencies (Fig 1D). To test this experimentally, we determined the encoding bandwidths of L4 and L5 neurons by analyzing the timing of their AP output during current noise injections. Injected currents were composed of small sinusoidal currents of various frequencies embedded in randomly fluctuating background noise with a correlation time $\tau$ of 5 ms (Fig 2F, Materials and methods). A small amount of constant DC current adjusted the mean membrane potential such that the fluctuating background drove cells to fire irregularly at about 5 Hz. The relation between AP timing and the embedded sinewave was determined for each frequency (Fig 2D and 2E). The raster plots and corresponding AP timing histograms demonstrate AP locking of L4 and L5 neurons to slower (50 Hz) and faster (200 Hz) sine wave frequencies. At 50 Hz, APs in both L4 and L5 neurons clustered to the depolarizing phase of the sine wave; at the higher frequency, phase-locking was much weaker in the L4 neurons (Fig 2D, 2E, and 2G). Statistical analysis revealed no significant difference in vector strength (Materials and methods) [6,31] at 50 and 100 Hz, whereas, at 200, 500, and 750 Hz, the vector strength was significantly lower in the L4 neurons (Fig 2G). These data confirm the theoretical expectation that the compact L4 neurons have a narrower encoding bandwidth than the L5 pyramidal cells with their extensive dendritic trees. However, actual spiny stellate neurons in vivo respond quite rapidly to weak afferent stimuli [32–34], and their functional bandwidth must actually be broader. We therefore tested the possibility that there is a difference in the other parameter that determines the encoding bandwidth. The actual background current of cortical neurons has never been experimentally measured.

## Fluctuating background current is slower in L4 neurons

During ongoing cortical activity, the background synaptic noise in a neuron reflects synaptic bombardment by asynchronous excitatory and inhibitory local inputs [4,32,33]. To preclude inputs from other brain areas and maximize the spatial effectiveness of the voltage clamp, we recorded in brain slices. Because neocortical neurons maintained in brain slices are largely quiescent, we induced reverberating synaptic activity by activating the network. This was accomplished by uncaging glutamate (MNI-glutamate) several hundred microns away from the recording site (Fig 3A and 3B). Care was taken to ensure that the uncaged glutamate did not come into direct contact with the cell under study, such that the subsequent synaptic barrage that resulted tens of milliseconds later reflected the activity of the local circuitry. Because we were interested in the effect of background noise characteristics on AP firing, we held the neuron at −50 mV, close to AP threshold. In these voltage clamp recordings, space clamp was maximized by replacing intracellular $K^+$ with $Cs^+$ while intracellular $Cl^-$ was maintained at physiological level. As shown in Fig 3B, UV flashes (50 ms) elicited subsequent network activity lasting up to 0.5 sec. The consequent synaptic noise consisted of a mixture of inhibitory and excitatory synaptic events (S3A Fig). However, since the voltage was held near the IPSC equilibrium potential, the recorded current was almost entirely excitatory (S3A and S3B Fig).

**Table 1. Properties of the excitatory cells studied in tangential slices (layer 4) and coronal slices (layer 5) of somatosensory cortex. Values represent mean and standard error, p-values are uncorrected results of t-tests.**

| Parameter Cell type | $V_{rest}$ (mV) | $\tau_{mem}$ (ms) | $R_{input}$ (MΩ) | $C_{mem}$ (pF) | AP Threshold (mV) | AP Amplitude (mV) | AP max. dV/dt (V/s) | AP min. dV/dt (V/s) | Bursting (%) |
|---|---|---|---|---|---|---|---|---|---|
| Layer 4 ($n$ = 12) | −74 ± 1.2 | 31.3 ± 4.0 | 450.4 ± 68.1 | 75 ± 7.5 | −44.0 ± 1.4 | 69.3 ± 2.9 | 310.4 ± 23.7 | 60.3 ± 2.4 | 0 |
| Layer 5 ($n$ = 13) | −66.8 ± 1.1 | 17.3 ± 1.2 | 90.1 ± 10.6 | 212.5 ± 20.5 | −43.5 ± 1.7 | 83.4 ± 2.5 | 367.8 ± 20.1 | 78.2 ± 3.8 | 46 |
| $p$ value | <0.001 | 0.002 | <0.001 | <0.001 | 0.83 | 0.0012 | 0.07665 | <0.001 | |

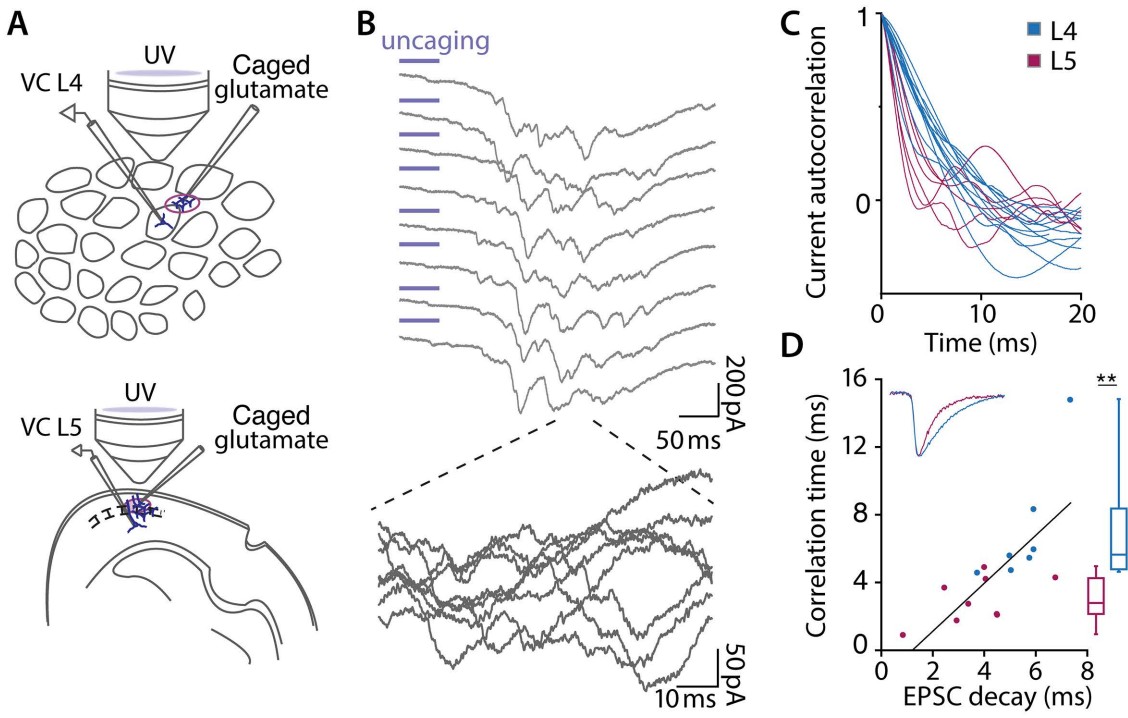

**Fig 3. Synaptic noise is slower in L4 neurons. (A)** Experimental setup for synaptic current noise recordings in vitro. Whole cell voltage clamp measurements in tangential slices (L4) and in coronal slices (L5). A second pipette carrying caged (MNI)-glutamate is situated 200−400 μm from the recorded cell and UV flashes (380 nm, 50 ms) release glutamate to activate the local synaptic network. **(B)** Current traces obtained in voltage clamp mode ($V_{hold}$ = −50 mV) showing synaptic fluctuations in response to 8 consecutive UV flashes. Note that each flash results in a burst of synaptic input with a waveform that differs from trial to trial. **(C)** Autocorrelation functions. **(D)** Correlation time of the synaptic noise plotted as a function of the decay time constant of spontaneous EPSCs. (Pearson Corr = 0.72, $n$ = 7 L4 cells and $n$ = 9 L5 cells for EPSC vs. correlation time). Box plots summarize the correlation times of synaptic noise (two-sided $t$ test, **$p$ < 0.01). Data https://doi.org/10.25625/VQUKFU.

We measured the temporal dynamics of the background activity in L4 neurons and found them to be significantly different than those of L5 neurons. The autocorrelation of the fluctuating background activity revealed that fluctuations in L4 decayed more slowly (Fig 3C). In Fig 3D, the plots were quantified by fitting a single exponential to the averaged autocorrelation function (10–20 trials per neuron); correlation time was 3–5 times longer in the L4 neurons. In order to determine how this difference in correlation time of the synaptic noise relates to the kinetics of the underlying synaptic events, we examined spontaneous excitatory post-synaptic currents (EPSCs) recorded at a holding potential of −70 mV, and inhibitory post-synaptic currents (IPSCs) at a holding potential of 0 mV in the two cell types (S3A and S3B Fig). While the time course of IPSCs in L4 and L5 cells was not significantly different, EPSC decay in L4 neurons was 2–3 times longer (S3C–S3F Fig). In addition, we found that the decay time constant of EPSCs was linearly related to the correlation time of the noise (Fig 3D). Inhibitory time constants have minimal, if any, effect (S3B Fig) because the measurements were made not far from the IPSC reversal potential (−60 mV).

## Slow background noise enables L4 neurons to code high frequencies

Our simple computational model suggested that a population of neurons with short dendrites could encode feedforward input better if the recurrent background input had a longer correlation time. We measured the correlation time in the small, L4 cells, and found it to be relatively long. We then determined how different background correlation times would affect the dynamic gain of the L4 cells. For the faster correlation time, we chose 2 ms, which is at the lower end of the correlation

PLOS Biology

times measured in L5 pyramidal cells. For the longer correlation time, we used 10 ms, slightly longer than the average 7 ms observed in L4 network reverberations. The raster plots and corresponding AP time histograms in Fig 4C and 4D compare the ability of 3 representative L4 neurons to follow a 200 Hz input signal when the noise was fast ($\tau = 2$ ms) and when the noise was slow (10 ms). It can be seen that the high-frequency tracking performance of the neurons was greatly enhanced with the slower correlation time. The vector-strength analysis for seven different input frequencies shows that slower background fluctuation is associated with a broadened bandwidth and a significant increase in the high-frequency encoding abilities of the L4 neurons (Fig 4G). Fig 4E shows that reanalysis of the recordings using the more efficient dynamic gain analysis (Supplementary material and S1A–S1F Fig) confirmed this result.

## Fidelity of population encoding depends both on the number of neurons and the background correlation time

Considering the theoretical predictions (Fig 1), we next sought to determine how large the L4 population must be for its output to reliably reflect brief, correlated changes in input. To this end, we embedded a single, time-locked artificial EPSC (aEPSC) in the random noise (as generated above), and determined how many sweeps are required to reliably detect

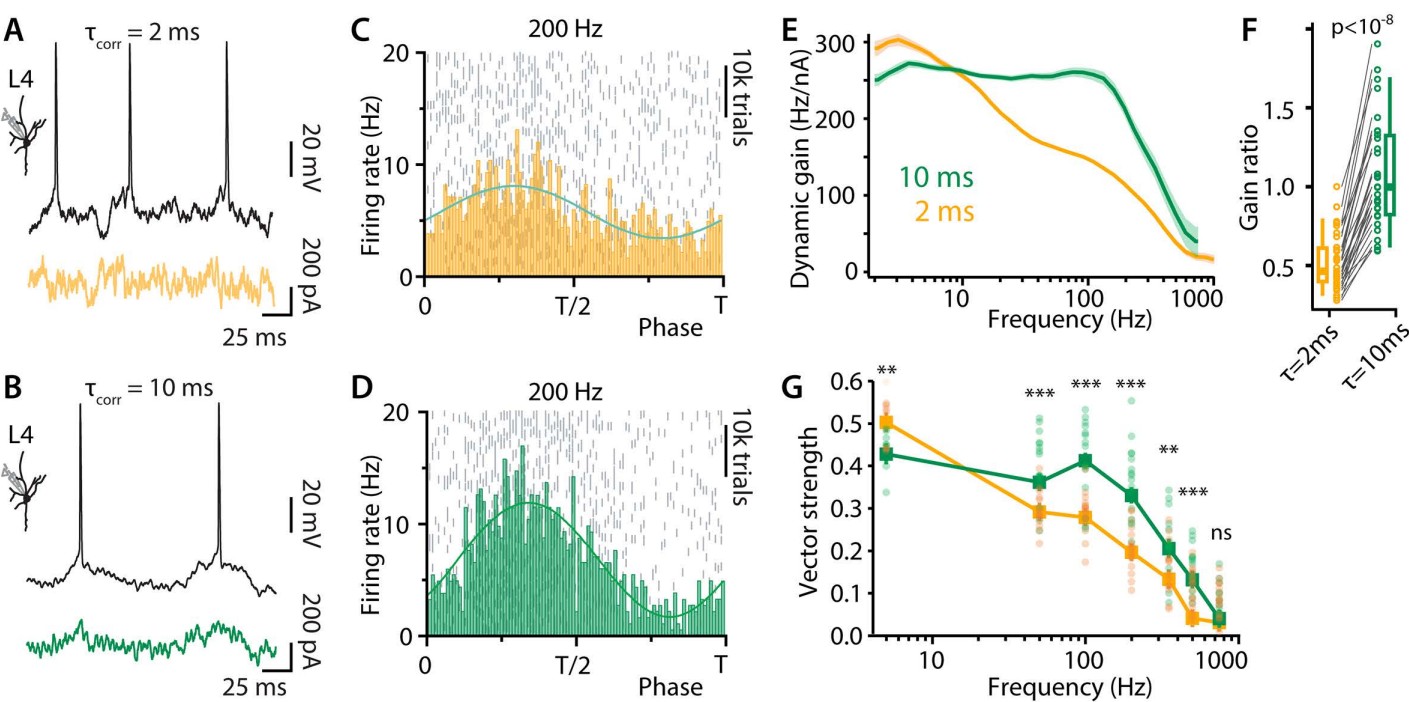

**Fig 4. Slow noise improves the encoding performance of L4 neurons.** Voltage fluctuations and neuronal firing in a L4 neuron in response to a 200 Hz sinusoidal modulation buried in synaptic noise with a correlation time of either 2 ms **(A)** or 10 ms **(B)**. Shown are brief sections of 45-second-long trials. **(C, D)** Raster plots and population rates showing the effect of the correlation time on the locking performance of the neurons to 200 (Hz) (APs were pooled from $n = 3$ L4 neurons). **(E)** Dynamic gain of L4 neurons as a function of input frequency. Pooled data from $n = 30$ L4 neurons are shown, each was stimulated with both correlation times. Solid lines and shaded areas represent the medians and confidence intervals of bootstrap statistics (see Materials and methods). **(F)** An individual dynamic gain curve $|G(f)|$ was calculated for each of the 30 neurons from **(E)**. Shown are the ratios $|G(100$ Hz$)|/|G(2$ Hz$)|$ for the two correlation times. Paired-data, two-sided Wilcoxon rank test indicated a significant difference between the data for the different correlation times with $p = 1.9e-9$. **(G)** Vector-strength as a function of sine wave frequency with fast and slow correlation times. Shown are results for individual trials (circles), medians (squares), and confidence intervals (vertical lines). The latter were calculated as described in Materials and methods. For each frequency, both correlation times (2 and 10 ms) were tested in the same cell. The significance of the effect of correlation time was determined with a paired, two-sided $t$ test. Sample size, uncorrected p-values and Cohen's D are, in sequence of increasing frequency: $n = 8$, $p = 0.003$, $D = 1.8$; $n = 15$, $p = 4e-4$, $D = 2.3$; $n = 16$, $p = 2e-4$, $D = 2.5$; $n = 16$, $p = 6e-5$, $D = 3.0$; $n = 14$, $p = 9e-4$, $D = 1.3$; $n = 16$, $p = 0.002$, $D = 1.5$; $n = 15$, $p = 0.89$, $D = 0.13$. Data DOI for (G) and (F): https://doi.org/10.25625/VQUKFU.

this small input in the output population spike train. In these experiments, since the noise is uncorrelated across sweeps, each sweep may be considered the equivalent of a different L4 neuron in response to the same aEPSC. The aEPSC (20 pA), which mimics the effect of a single thalamocortical AP [35,36] was superimposed on either 'slow' or 'fast' background fluctuations and injected into single cells every 50 ms. As can be seen in the traces in Fig 5A, AP clustering just after the aEPSC was most prominent when the background noise was "slow." This is a direct consequence of the wider bandwidth of the dynamic gain; in fact, the waveform and amplitude of the population rate can be quantitatively predicted from the dynamic gain (S1G–S1J Fig). In the peri-stimulus time histograms of Fig 5B and 5C, it can be seen that for an essentially "infinite" population (n = 25,200), a response to the aEPSC is readily detected at both slow and fast noise correlation times. However, when the population consists of fewer neurons, the aEPSC is much harder to detect with the faster time correlation of the noise. Analysis of the detection probability as a function of the population size (see Materials and methods) reveals that 'slow' fluctuations improve the reliability of detection by 70%−100% for firing rate changes occurring within 1 ms (Fig 5D). There are a few thousand excitatory neurons in L4 of a single barrel [20]. Because in vivo a thalamocortical volley is immediately followed by strong feedforward and feedback inhibition [37,38], our data indicate that the slower correlation time of the noise is essential for the fast encoding that is actually achieved by L4. Note that in spiny stellate cells in vivo, the background noise largely reflects recurrent activity, and the histogram bars after the initial response might be modified as a consequence of the thalamocortical EPSC. However, only the first one or two bars are relevant for our conclusion (Fig 5B–D).

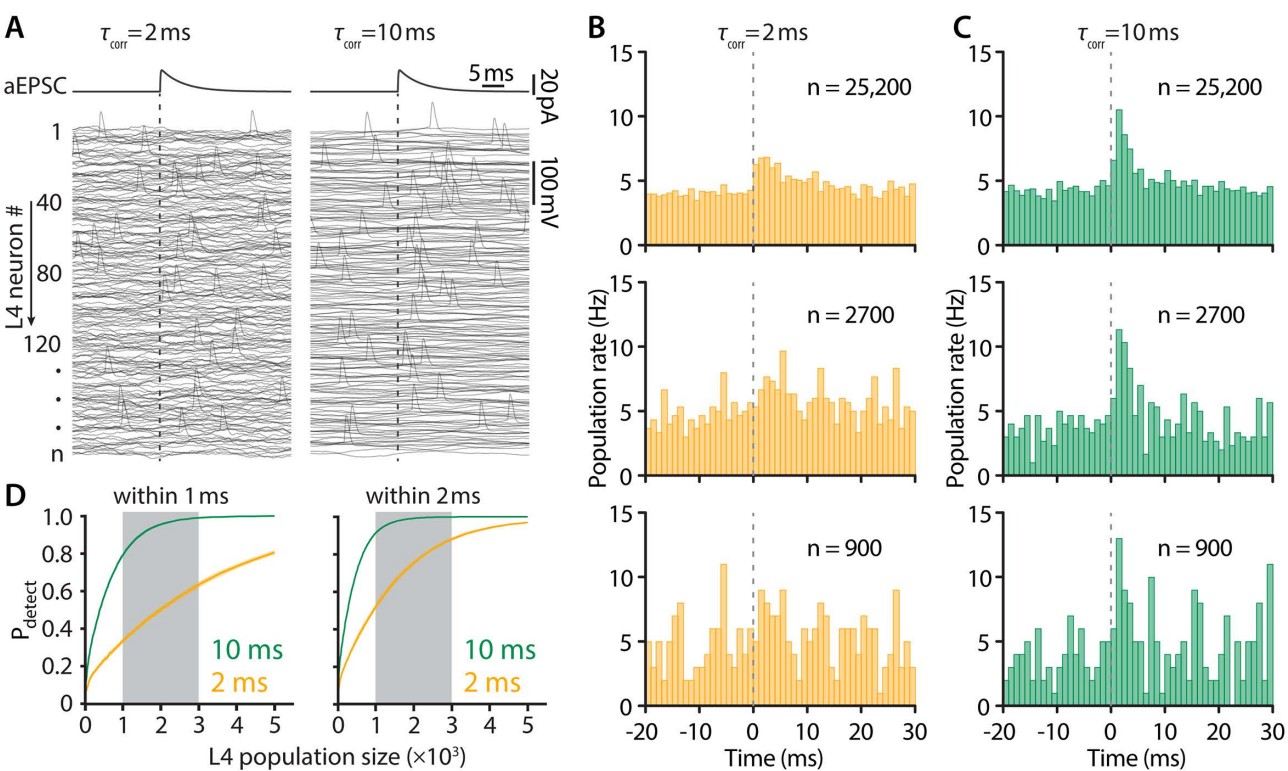

**Fig 5. Slow background noise sharpens the L4 population response. (A)** A single artificial EPSC, embedded in either fast (left) or slow (right) noise, was injected into a population of 'n' L4 excitatory neurons. Example voltage waveforms are shown. **(B)** and **(C)** Example peri-stimulus time histograms for population size "N" of 25,200, 2,700, and 900. **(D)** The effect of synaptic noise correlation time on detection probability of input within 1 ms and within 2 ms. (Median + shaded 95% confidence interval from bootstrap). Data are pooled from n = 13 L4 neurons.

## Fast encoding depends on K$_V$7 potassium currents

Our experiments show that when background noise fluctuates slowly, the population of L4 neurons rapidly encodes thalamocortical input volleys. This is in agreement with predictions of our simple theoretical model in which AP generation occurs instantaneously (Fig 1). However, in real neurons, AP generation speed is finite and limited by the sodium and potassium currents that underlie AP initiation [13,39,40]. In the axon initial segment (AIS), sodium channels and K$_V$7.2/3 potassium channels are arranged in a highly specific organization [41], and these K$_V$7.2/3 channels modulate the AP initiation [42,43]. We sought to determine whether these specific potassium channels may regulate the dynamic gain of the L4 neurons by repeating the experiments from Fig 4 while blocking K$_V$7 channels with 20 µM of the specific antagonist, XE991 [44]. In Fig 6A, comparison of average AP waveforms during 10 ms and 2 ms background activity with and without application of the K$_V$7 blocker demonstrates an effect of the potassium current on the voltage dynamics leading up to the AP. Under control conditions, the slower background was associated with a far more gradual voltage incline to AP threshold than the faster background. This difference is nearly abolished by XE991. The finding is quantified in Fig 6B, which plots the difference between the voltages recorded 5 ms before AP initiation for the two background correlation times. Dynamic gain analysis shows that the addition of XE991 to the bath completely abolished the expanded bandwidth of L4 neurons despite the characteristically slow background noise. Thus, blockade of these K channels removed the ability of the L4 neurons to faithfully respond to a brief thalamocortical volley. Because K$_V$7 channels underlie the acetylcholine-sensitive M-current [45], and acetylcholine is a prominent neuromodulator in the cortex [46], we propose that the coding ability of L4 may vary as a function of brain state.

## Discussion

We developed a simple model, showing that a neuronal population encodes feedforward input more faithfully, if the population is numerous, its neurons' dendrites are large, or the recurrent input is more slowly fluctuating. The smallest excitatory cortical neurons are spiny stellate cells that form a key gateway of sensory input into the cortex. We show that in layer 4 of the somatosensory cortex, the temporal properties of background noise, axonal potassium channels, and neuronal population size, offset the small size of the excitatory neurons, so as to assure faithful transmission of even the weakest thalamocortical input. This study provides a direct experimental assessment of information encoding in a

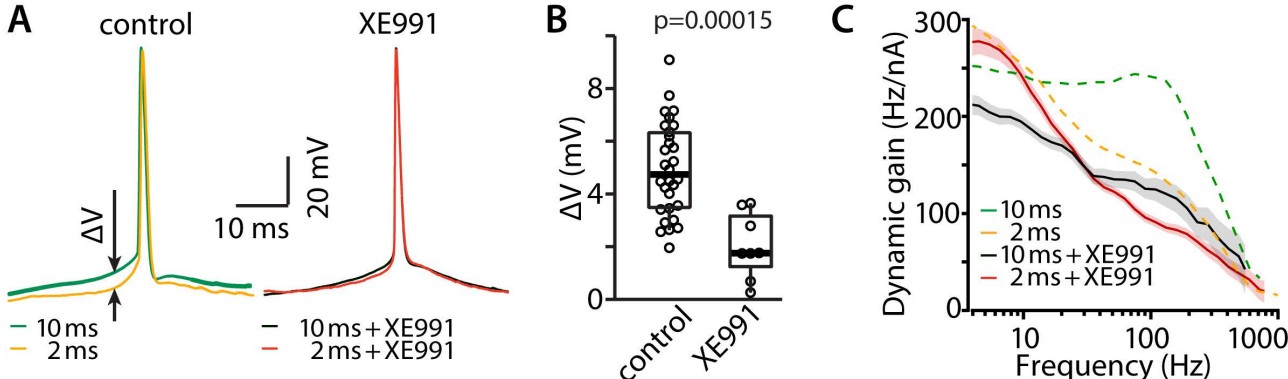

**Fig 6. K$_V$7 activity is required for more precise AP firing under slow background correlations. (A)** Average AP waveforms of one neuron in control conditions (left) and one neuron under XE991 application (right). The membrane voltage 5 milliseconds before AP threshold differs in fast and slow background noise (ΔV, arrows). **(B)** The voltage difference, defined in **(A)**, is shown for the 30 cells from Fig 4 (control) and 8 cells under XE991 block. $p = 0.00015$, two-sided Wilcoxon rank test. Data https://doi.org/10.25625/VQUKFU **(C)** Slow background improves the dynamic gain bandwidth when K$_V$7 channels are active (orange to green from Fig 4E). This 'high-frequency boost' is largely abolished, when Kv7 channels are blocked (red to black).

population of cortical neurons using theoretical concepts proposed 50 years ago [7] and, for the first time, relates dynamic gain to a specific cortical function.

We demonstrate that (1) L4 neurons generate synaptic noise of very slow correlation time, which increases their encoding bandwidth, (2) broad bandwidth enables populations of only hundreds of neurons to reliably reflect single thalamocortical APs with millisecond precision, and (3) changes in L4 neuron properties that appear to have only a minor influence at the single cell level, strongly impact coding by the population as a whole. Although our study is focused on layer 4 of the rodent barrel cortex, we expect that our findings are broadly applicable to the mechanisms of cortical neuronal population coding in general. For example, it has been shown that a single externally induced AP can lead to a detectable increase in the activity of a population over several dozen milliseconds [47] and can even have a behavioral consequence [48].

Almost all neurons in an S1 cortical column receive direct thalamocortical input. But while all other cells receive context information from other whiskers, from secondary somatosensory cortex and primary motor cortex [49], the connectivity of spiny stellate cells is uniquely focused. Their only external excitatory input originates in the specific thalamus [17]. Thus, the output of the L4 neuronal population, which projects throughout the column [17], constitutes a beam of exclusively sensory information. Strong feedforward and feedback inhibition leaves only a narrow window of about 2 ms during which these cells must respond [37,50,51], such that the relayed output from L4, by amplifying the thalamic input, serves to focus the activity of the entire cortical column. Our findings explain how these neurons, despite their small size, are able to respond within this extremely brief window. Because the correlation time of the background noise is slow in L4 (Fig 3), less than 1,000 L4 neurons are sufficient to allow detection of a single thalamic EPSC with 90% confidence within the time constraints imposed by inhibition (Fig 5). Remarkably, this corresponds well to the number of L4 neurons (700) contacted by a single thalamocortical axon [52].

We provide the first direct measurements of the kinetics of background current noise and find that, whereas it is fast in the L5 neurons, it is slow in the much smaller L4 neurons (Fig 3). We further demonstrate that, as theoretically predicted, the slow kinetics of the noise is very likely due to long EPSC decay time constants. This difference between L4 and other layers may be explained by previous evidence that, although fast AMPA-receptors are the predominant mediators of excitatory synaptic activity in all other cortical layers, in L4 of the mouse barrel cortex, a slower NMDA-receptor with minimal $Mg^{2+}$-sensitivity (NR2C) is mainly responsible for ongoing synaptic activity, even at resting potential [27,53]. Because encoding capacity critically depends on background correlation time (Fig 4), the unique synaptic physiology in L4 is crucial to its function.

An alternative mechanism for slow input fluctuations could be a slow oscillation of the recurrent network itself. Although this would assure conditions for a rapid response of L4 neurons without the need for special synaptic receptors, those oscillations would pose a serious problem for L4 function because it would create prolonged periods during which most neurons were far from threshold, and the population's input sensitivity would be reduced. Indeed, in vivo recordings from mouse primary somatosensory cortex (S1) show that during active sensing periods, the networks in S1 switch to asynchronous activity [33], thereby avoiding intermittent drop-outs. The input correlation time is at least as long as the decay time constant of the dominant post-synaptic currents, even if the network activity is completely asynchronous. The presence of slow synapses, thereby, allows L4 neurons to consistently respond rapidly.

Ultimately, the bandwidth of dynamic gain must reflect the ionic mechanisms underlying AP generation [54]. Briefly, wide bandwidth requires that the voltage dependence of inward current at threshold be steep [54]. In cortical neurons, APs are generated in the proximal part of the axon, the AIS, where an elaborate cytoskeleton anchors both voltage-dependent Na channels and voltage-dependent $K_V7$ channels in a precise spatial pattern [55]. We suggest that, because the slower background noise is associated with a much more gradual incline of the pre-AP membrane potential towards threshold, the local K channels are activated, the Na current is opposed, and by the time threshold is reached the membrane voltage is at a much steeper point on the Na activation curve. This would explain our finding that blocking the $K_V7$ channels, by removing the difference in pre-AP voltage trajectory, narrowed the bandwidth. This result predicts that the relay function of

the L4 population may vary as a function of brain state, since $K_V7$ channels are regulated by the neuromodulator acetylcholine [56].

Fundamentally, cortical function entails the passage of information between neuronal ensembles composed of neurons whose output collectively represents common input at a given point in time. However, a given neuron joins different populations at different moments, and the specific function that is served is usually not known. By studying this encoding in a well-defined subpopulation with known inputs, we identified principles of encoding that may well apply to many other cortical areas. By measuring the bandwidth of the dynamic gain of a single neuron, we obtain insight into how this neuron and others like it participate in the dynamics of cortical function. We note that the dynamic gain of a given neuron is not necessarily a fixed parameter, but may be sensitive to physiological or pathophysiological states. Merino and colleagues [57] reported that in L2/3, when slow oscillatory brain rhythms appear, they are reflected in the background noise, and the dynamic gain function of certain inhibitory neurons is thereby dramatically broadened. Revah and colleagues [40] showed that dynamic gain is modified due to changes in the AP initiation machinery during recovery from brief epochs of transient cortical anoxia and consequent spreading depolarization. The findings illustrated in Fig 6 suggest that the dynamic gain of some cortical neurons may be modified during normal behaviors that are associated with changes in $K_V7$ potassium channel activation.

It is widely agreed that the purely sensory, whisker-specific input to spiny stellate cells establishes L4 as the key entrance for sensory input to cortex. Our characterization extends this view. We found that three parameters from completely different domains, i.e., population size, neuron morphology, and NMDA receptor kinetics, act together to set the observed sensitivity of L4 excitatory neurons. This allows L4 to relay the thalamic input rapidly and amplify it. At the same time, our preliminary characterization of a possible cholinergic modulation suggests that L4 may even serve as a gate, operated by neuromodulation. This exciting possibility inspires further characterization of cholinergic modulation ex vivo and, importantly, in vivo, to identify behaviorally relevant effects of cholinergic modulation on sensory performance, particularly for weak tactile stimuli, when amplification by L4 might be particularly important.

## Materials and methods

### Ethics statement

All animal procedures were conducted in accordance with the Prevention of Cruelty to Animals (Experiments on Animals) Law, 5,754–1994 (Israel). All experiments were approved by the Animal Care and Use Committee of the Hebrew University of Jerusalem (permit number: MD-13-13607-2).

### Animals

ICR (CD-1) mice were maintained in a specific-pathogen-free facility at the Hebrew University of Jerusalem, Rehovot, under veterinary supervision. Animals had ad libitum access to food and water and were housed under a 12 h light/dark cycle (lights on at 7:00 A.M.). Three- to four-week-old mice (20–30 g) were housed in groups of 5–10 per cage. Both male and female mice were used in this study. Mice were anesthetized with pentobarbitone and euthanized by cervical decapitation. All efforts were made to minimize pain and stress.

### Electrophysiology

Recordings were made in 300−400 µm thick coronal slices (L5) and tangential slices (L4) of the somatosensory cortex of CD-1 mice of either sex that were 14–24 postnatal days of age. Procedures for the preparation and maintenance of slices were similar to those described previously [58]. Animals of either sex were deeply anesthetized with Nembutal (60 mg/kg) or isoflurane and killed by decapitation; their brains were rapidly removed and placed in cold (4°C), oxygenated (95% $O_2$_5% $CO_2$) artificial cerebrospinal fluid (aCSF). Coronal and tangential slices from a region corresponding to the primary

somatosensory cortex were cut on a vibratome (Series 1000; Pelco International, Redding, Canada), as previously shown [27]. The slices were then placed in a holding chamber containing aCSF at room temperature for >30 min of recovery and transferred to the recording chamber for recordings and analysis.

Whole-cell patch-clamp recordings were made from visually identified L4 and L5 neurons under infrared differential interference contrast (IR-DIC) microscopic control. Slices were held submerged in a chamber on a fixed stage of an Axioskop FS microscope (Carl Zeiss, Oberkochen, Germany). Voltage and current were recorded in whole-cell configuration using an Axoclamp 2B (Molecular Devices, Foster City, CA) connected to an MU-type head stage (Molecular Devices, Foster City, CA). Patch pipettes were manufactured from thick-walled borosilicate glass capillaries (outer diameter, 1.5 mm; Hilgenberg, Malsfeld, Germany) and had resistances of 5–7 MΩ for somatic and extracellular recordings. All recordings were made at 30±2°C. Command voltage protocols were generated, and whole-cell data were acquired on-line with a Digidata 1320A analog-to-digital interface. Data were low-pass filtered (−3 dB, one-pole Butterworth filter) and digitized at 20 kHz, except for spike shape analysis, which was oversampled at 250 kHz. For each neuron, bridge correction and capacitance compensation were performed manually online.

The aCSF contained (in mM): 124 NaCl, 3 KCl, 2 CaCl2, 2 $MgSO_4$, 1.25 $NaH_2PO_4$, 26 $NaHCO_3$, and 10 glucose, pH 7.3 when bubbled with a 95% $O_2$—5% $CO_2$ mixture. The pipette solution for current clamp experiments contained (in mM): 130 K-gluconate, 6 KCl, 4 NaCl, 2 $MgCl_2$, and 10 HEPES (potassium salt), pH 7.3. For voltage clamp recordings the solution contained (in mM): 135 CsOH, 124 gluconic acid, 6 CsCl, 2 $MgCl_2$, 4 NaCl, and 10 HEPES (potassium salt), pH 7.3. The intracellular chloride concentration is the same for both recipes, resulting in a calculated reversal potential for chloride of −59.7 mV.

**Assessing passive and active membrane properties.** The subthreshold membrane properties and the characteristics of the AP were obtained by injecting a series of increasing 500 ms current steps, separated by 2–5 s intervals during which the cell was held at either resting or at −77 mV.

**Synaptic noise simulation in brain slices.** To achieve periods of reverberating network activity, in addition to the recording electrode, a second borosilicate glass electrode, pulled to a resistance of 2–4 MΩ, containing caged glutamate (MNI-caged-L-glutamate, 10 mM, Tocris), and connected to a Picospritzer II device (PARKER HANNIFIN CORPORATION), was placed 100–200 μm away from the recorded neuron. Caged glutamate was puffed locally for 1 second, and, following, a short, 50–100 ms long UV light-pulse was delivered onto the tissue through a 40× objective (~25 mW/mm²). Successful network activations resulted in incoming synaptic barrages impinging on the recorded cell after a considerable delay (see Fig 3). Special care was taken to prevent glutamate from directly exciting the cell. Specifically, we repeatedly checked that the uncaging UV focus was outside the boundary of the barrel we recorded from. Because the dendrites of L4 neurons are almost completely confined to a barrel, this largely excludes the chance of direct contact of the released glutamate with the stimulated neuron's dendrites. In coronal slices, we performed release at least 200 μm away from the recorded Layer 5 neuron soma. Successful separation of uncaging and reverberation detection was indicated by a delay of at least 25 ms from the UV pulse onset.

## Assessing the frequency response function of L4 and L5 neurons

To assess the frequency response of neuronal populations, we injected stochastically fluctuating currents in the whole-cell configuration. This analysis aims to achieve an in vivo-like operating point, mimicking a situation in which a high rate of synaptic inputs provides a continuously changing net background current, and AP firing is driven not by the average input but by its transient depolarizing excursions. The injected currents were composed of a stochastic signal, and a sinusoid signal of frequency $f_s$ (between 5 and 750 Hz as indicated). The stochastic signal was based on an Ornstein–Uhlenbeck (OU) process, i.e., low-pass filtered white noise with a Gaussian amplitude distribution. Its correlation time was either 2, 5, or 10 ms as indicated. While 5 ms was chosen because this was previously assumed to be representative of glutamatergic synaptic decay time constants, the 2 and 10 ms were chosen based on our measurements (Fig 3). There is no guarantee

that input correlations in vivo follow the exact same pattern. Although the PSC decay time constants represent a lower bound for the input current correlation time, multiple processes may prolong it. Our choice of 10 ms slightly exceeds the 7 ms we observed in L4 network reverberations. This experimental design thereby accommodates, to some degree, possibly slower correlations in vivo. We adjusted the input's standard deviation, $\sigma_I$, based on the input resistance of the neuron, to ensure that the voltage fluctuations were similar in all neurons. Along with $\sigma_I$, we also scaled the amplitude of the sinusoidal component $A_{sin} = 0.37 \cdot \sigma_I$. A constant, direct current (DC) $\langle I \rangle$ was added and adjusted to maintain a target firing rate of ~5 Hz. In the absence of measurements of spiny stellate cells' firing rates in vivo, we opted for this value because the neurons were able to sustain it for many seconds without detectable changes in AP shape. Lower values would inversely prolong the required measurement duration and preclude the application of two different input statistics to the same neuron. Currents were injected in 46 s episodes, with ~30 s intervals between the injections. APs were detected as upward crossings of zero membrane voltage.

**Vector strength.** A neuron's ability to encode signals can be evaluated with different methods. We used two methods to ensure precise results. Results for both are obtained with code available at https://github.com/Anneef/AnTools in the subfolder 'Dynamic Gain Code'. The permanent version of the code is published here: https://doi.org/10.25625/VQUKFU.

The first method relies on the presence of a sinusoidal component in the input and uses the vector strength $r$ to characterize how the AP times $t_j$ lock to the phase of the periodic stimulus:

$$r = \text{abs}\left(\sum_{j=1}^{N} \exp\left(i \cdot 2\pi \cdot f \cdot t_j\right)\right) / N$$

(1)

In the experiments underlying Fig 2, where stimuli of a single correlation time were used (5 ms), each cell ($n = 10$ L4 cells, $n = 9$ L5 cells) was typically studied with all 5 frequencies ($f = 50, 100, 200, 500,$ and 750 Hz) twice (for L4 cells) or thrice (for L5 cells). In the experiments underlying Fig 4, 30 L4 neurons received fluctuating stimuli of 2 and of 10 ms correlation time. For both correlation times, typically, 3–4 of the 7 frequencies (5, 50, 100, 200, 350, 500, and 750 Hz) were applied once to each cell. For the number of stimuli applied, the cell in S1A–S1F Fig is not typical because, in this cell, a total of 11 trials were successfully recorded. In terms of results, it is a typical cell. Out of the 204 trials recorded from these 30 cells, 200 trials were part of a 2–10 ms pair, i.e., the trial stimuli contained the same sine frequency for both correlation times, recorded in the same cell. In Fig 4F, the vector strengths calculated for each of these 200 trials are shown in circles. The average vector strengths, calculated for all trials with a given sine frequency, are shown as squares.

Note that the average values are not simply the averages of the individual vector strength values because complex numbers are averaged and only their absolute values are reported. Specifically, the average absolute value is at most equal to the average of the individual absolute values of the trials. This case occurs only if the phases of all trial vector strengths are identical, i.e., if $r_{trial} = \frac{1}{N_{trial}} \sum_{j=1}^{N_{trial}} \left(2\pi \cdot f \cdot t_j\right)$ is the same for all trials.

Furthermore, note that each of the 204 trials contributes to the calculation of the dynamic gain in Fig 4E.

To determine the confidence intervals and statistical significance of phase locking (vector strength $r$) of recorded APs to periodic input current stimulation, we used 10,000 bootstrap samples of the actual spike times and of randomized spike times similar to the procedure described below for the dynamic gain. Because the contributions of individual APs are points on the unit circle, the average vector of the kth bootstrap sample is a point somewhere inside that unit circle, characterized by its magnitude and phase: $m_k \cdot e^{i\phi_k}$. To obtain the confidence interval of the magnitudes, it is not correct to simply study the distribution of all $m_k$. Instead, the bootstrap results first have to be projected onto the vector of the average $\frac{1}{N} \sum_{j=1}^{N} \exp(i \cdot 2\pi \cdot f \cdot t_j) = m_{avg} \cdot e^{i\phi_{avg}}$. The central 95 percentiles of these projected vectors' magnitudes, i.e., of $m_k \cdot \cos(\phi_k - \phi_{avg})$, corresponds to the confidence interval of the vector strength as shown in Figs 2, 4, and S1 Fig.

**Dynamic gain calculation.** The second method to characterize the encoding ability of neurons is the dynamic gain function $G(f)$. It represents the frequency-domain, linear transfer function (or susceptibility) of a population of neurons that receive a common feedforward input $\Delta I$. In other words, in the frequency domain, the neurons' common population rate changes as follows:

$$\Delta \widetilde{\nu}(f) = \widetilde{I}(f) \cdot G(f) \tag{2}$$

The tilde represents the Fourier transformations of the respective functions.

The calculation of the dynamic gain, as performed here, based on [59], does not rely on an embedded sinusoidal signal. Instead, all the frequency components that are present in the stochastic signal are used as reference to calculate phase-locking. We described the rationale and calculations in detail earlier [13,57]. A spike-triggered average (STA) input current was obtained by summing up 1-s-long stimulus segments centered on the AP times for all cells of a given condition and dividing by the total number of APs.

The complex dynamic gain function $G(f)$ was calculated as the ratio of the complex conjugate of the Fourier transform of the STA, $F(STA_I)^*$, and the Fourier transform of the autocorrelation of the stimulus (its power spectrum), multiplied with the average firing rate:

$$G(f) = \frac{F\left(STA_I\right)^*}{F(AC_I)} \langle \nu \rangle \tag{3}$$

In order to improve the signal-to-noise ratio, $G(f)$ was filtered in the frequency domain by a Gaussian filter $w(f')$ centered at frequency $f' = f$ and a frequency-dependent window size with a standard deviation of $f/2\pi$ as proposed by [59]:

$$w\left(f'\right) = \frac{1}{\sqrt{2\pi}\left(\frac{f}{2\pi}\right)} exp \left[ \frac{-1}{2}\left(\frac{f'-f}{\frac{f}{2\pi}}\right)^2 \right] \tag{4}$$

The filtered dynamic gain function $G_w(f)$ thus becomes

$$G_w(f) = \frac{\int G\left(f'\right) \cdot w\left(f'\right) \cdot df'}{\int w\left(f'\right) \cdot df'} \tag{5}$$

The magnitude $|G|$ of this complex dynamic gain function is shown in Figs 4 and 6. Its phase $arg(\langle G(f) \rangle)$ is not displayed because it is not relevant for the arguments here. To determine the region of the gain curves, where $|G|$ is significantly larger than zero, a noise floor was calculated as the 95th percentile of the bootstrap distribution of dynamic gain functions obtained from AP times for which the temporal relation to the stimulus is destroyed by shifting the original times by an interval larger than 1 second. The noise floors are shown in Fig 4E as dashed lines. The results are shown only for frequencies where $|G|$ is above the noise floor. In accordance with the calculation of the confidence interval of the vector-strength-based gain, again, the confidence interval of $|G|$ is obtained by bootstrap-sampling over all AP times, and as before, the resulting complex bootstrap values $G(f)$ are projected onto the angle of the average $\langle G(f) \rangle$, before the central 95% of the $\left| G(f) \right|$ were determined. As explained above, the projection ensures that the confidence interval includes zero at those frequencies where the gain exceeds the result for randomized spike times (noise floor). We used 1,000 bootstrap samples to estimate noise floor and confidence interval.

**Comparing calculations of dynamic gain based on vector-strength and spike-triggered average.** In S1A–S1F Fig, the two methods are compared. As long as the exact same data are analyzed, i.e., a single trial with one underlying

sine-frequency, the results of the two methods are in good quantitative agreement and statistically indistinguishable. It should be noted, however, that an entire curve for various frequencies represents different information for the vector strength method and the STA-based method. Whereas the latter uses the same data (all APs in all trials) for all data points, the former calculates the gain at a given frequency only from APs fired within the one (or few) trials, for which a sinusoidal stimulus component with this particular frequency was added to the OU input. So unless multiple sine components are added to the same trial, all points in the vector-strength-based dynamic gain curve originate from disjunct datasets.

The analysis in S1A–S1E Fig highlights the great advantage of the STA-based method: each AP is analyzed with respect to all input frequencies and not just a single frequency of a superimposed periodic input component. For correlation times as short as 2 ms, the input contains so much power throughout the interesting frequency range (1–800 Hz), that the $STA_I$-based dynamic gain analysis provides confidence intervals as narrow as the vector-strength method for all relevant frequencies. For such short input correlation times, usage of the vector strength method seems to offer no advantage. In the case of longer input correlations (10 ms), the confidence intervals from the STA-based dynamic gain curve are very wide for high frequencies. For the low number of spikes fired during a single trial, the vector strength provides a more precise estimate, because at the one frequency of interest, the underlying OU process contains very little power and therefore limits the signal-to-noise of the dynamic gain at this frequency. However, as soon as vector strength measurements at more than three or four frequencies are conducted, the APs from all these trials together allow a precise STA-based estimate of the gain at all frequencies and once more the vector strength is outperformed.

**Dynamic gain accurately describes the rate change in response to an artificial EPSC.** Here, we provide detailed technical descriptions of the mathematical connections among the presented results. In particular, we show how the frequency-domain expression of the dynamic gain function $G(f)$ is related to the temporal response, i.e., the change of the firing rate over time $\Delta\nu(t)$ after the input has changed with $\Delta I(t)$. We introduced the dynamic gain as the linear transfer function of a population of neurons, with respect to changes in the common (feedforward) input. This is expressed in the frequency domain through Eq. 2. Its time-domain equivalent is a convolution:

$$\Delta\nu(t) = \int_{-\infty}^{t} \Delta I(\tau) \cdot \widetilde{G}(t-\tau)d\tau$$

(6)

The frequency domain equivalent is a simple multiplication:

$$\Delta\nu(t) = \mathcal{F}^{-1}\left(\mathcal{F}\left(\Delta I(t)\right) \cdot G(f)\right)$$

(7)

It is not a priori clear, to which degree this linear approximation captures the population's response to a change in the common input $\Delta I$. For very small perturbations, the linearity can be plausibly assumed, but what input magnitude can be considered sufficiently small? We can test this for a concrete realization of $\Delta I$ by comparing the predictions of the time-domain equivalent of Eq. 2 and the experimentally obtained $\Delta\nu$.

This test is conducted in S1 Fig. We obtained the dynamic gain curves from APs fired at least 25 ms after the EPSC onset (S1G Fig). Notably, those dynamic gain curves are very similar to the gain curves in Fig 4, recorded from cells in the same cortical area and with similar cellular properties. This similarity indicates that the presence of the aEPSC signal did not substantially influence the dynamic gain calculation. In the next step, we took those complex dynamic gain functions (including the phase information) and, following Eq. 7, we multiplied $G(f)$ with the Fourier transformation of the waveform $\Delta I(t)$ of the aEPSCs (Figs S1F and 5A), the inverse Fourier-transformation results in a prediction for the response $\Delta\nu(t)$, shown in S1H Fig. Finally, we compared this theoretical prediction with the experimental results that are shown in the top panels of Fig 5B and 5C. This comparison demonstrates that the theoretically predicted responses capture all features of the experimental responses, i.e., delay, magnitude, and temporal extent (S1I Fig). Importantly, the different dynamic

gain curves for background correlations of 2 and 10 ms closely predict the different response waveforms in Fig 5B and 5C, respectively. This demonstrates that the dynamic gain, the linear approximation of encoding, faithfully describes the encoding in the conditions tested, i.e., at *a given working point* characterized by input parameters $\mu_I$, $\sigma_I$, $\tau$, and AP firing statistics. That notwithstanding, it is obvious that the system is not *globally* linear. This is evident from the very fact that the temporal statistics of the input has a profound impact on the dynamic gain. This would not be the case, if the system was globally linear.

## Theoretical model

For the neuron model, we had to find a possibility to study the effect of dendrite properties and input correlation times. We implemented a simple Gauss neuron, in which AP firing is not explicitly modeled, but instead each crossing of a voltage threshold from below is detected as an AP. From our earlier characterization of this model, we knew, that its encoding ability depends on the correlation time $\tau$ of the input current [60].

The electrotonic structure, i.e., the passive properties of the neuron model and its morphology, determine how the current input is transformed into the fluctuating membrane voltage. If the model was only a point neuron with resistance R and capacitance C, the transformation would correspond to a low-pass filter $\widetilde{V}(\omega) = R \cdot \widetilde{I}(\omega)/\sqrt{(1 + R^2C^2\omega^2 \cdot)}$. When a dendrite is added, low frequencies experience an additional suppression, while very high frequencies are less affected. For the transfer function for currents injected into the soma of this more complex structure (neuron + dendrite), we used an analytical expression derived in [61]. The soma was represented by a sphere with 10 μm diameter, the dendrite is a cylinder with 5 μm diameter and variable length (as specified in Fig 1). Intracellular and membrane resistivity were 1.5 Ωm and 0.9 Ωm², the specific membrane capacitance was 0.01 F/m². For reference, these parameters result in a membrane time constant of 9 ms and an electrotonic length constant of 866 μm.

## Statistical tests

Before tests were applied, the individual distributions were tested against the normal distribution with Jarque–Bera tests. If the hypothesis of a normal distribution could not be rejected, the hypothesis of equal variances was tested with *F*-tests. If not rejected, the distributions were then compared with *t*-tests. In Figs 4F and 6B, those criteria were not passed, and therefore a Wilcoxon rank test was used.

**Optimal decoder and detection probability.** The optimal decoder is set up as described earlier [6]. It monitors the number of APs $n_{detect}$ fired by a neuronal population during a "measurement window" of a certain width (3 ms Fig 1; 1 and 2 ms in Fig 5D left and right panel). If the AP count in this window exceeds the threshold $n_{thresh}$, defined as the 95th percentile of the distribution of AP counts in all the baseline windows (of equal width), then a significant increase is detected.

Because $n_{detect}$ and $n_{thresh}$ are both discrete random numbers with distributions following count statistics, the detection probability shows abrupt jumps as the population size increases. Every time n$_{thresh}$ is incremented by one, the detection probability drops. These jumps are artifacts of the decoder definition. They overlay the steady increase of the underlying detection probability with population size. We therefore do not show these jumps but only the overall trend of increasing P.

## Supporting information

**S1 Table Stimulus details for the recordings underlying Fig 2.**
(PDF)

**S2 Table Stimulus details for the recordings underlying Fig 4.**
(PDF)

**S1 Fig. Linear ansatz to quantify encoding is confirmed by comparison of three different approaches. (A)** In all cases, the neurons' working points are set by fluctuating inputs with a correlation time of either 2 ms (orange) or 10 ms (green). Mean and standard deviation are chosen to achieve irregular firing with an average rate $\langle v \rangle$ around 5 Hz. (see Fig 4A). This stochastic background stimulus (Ornstein–Uhlenbeck-Type, see Materials and methods), can be combined with a periodic, sinusoidal input. Here, examples are shown with frequencies $f_s$ of 50 Hz (black) and 350 Hz (gray). **(B)** When AP times are evaluated only with respect to the phase of the added sine component, the sine-locked firing rate is obtained (bars). It follows the sinusoidal input modulation with a phase shift and a modulation depth amplitude $A_v$. Each of the two panels displays data of a single, 45-second-long stimulation trial, evoking about 220 APs. **(C)** From all AP times $t_j$ the vector strength can be calculated. The results for the two example trials from B are shown in the left panel. The right panel contains the results for all 11 trials in this cell (5 different sine frequencies for 2 ms correlated background and 6 frequencies for 10 ms). Shown are average and 95% confidence intervals. **(D)** AP times can also be evaluated with respect to all the frequency components that jointly constitute the noisy background and the sine component. To do so, the input auto-correlation $AC_I$ and the spike-triggered average input $STA_I$ were obtained from the total input and the times $t_j$, at which action potentials occurred. Shown are results for the 11 individual trials (semi-transparent) and for the average of the trials with the same correlation time (opaque). Note that the $AC_I$ of each trial clearly displays a periodic component, reflecting the sinusoidal current that had been added to the stimulus (see A). **(E)** The dynamic gain function G(f) can be calculated as the ratio of the Fourier-transformations of $STA_I$ and $AC_I$, multiplied with the average firing rate. The left panel shows its magnitude |G| for two individual trials, for correlation times of 2 (orange) and 10 (green) milliseconds. Note how already single trials reveal the strong dependence on correlation time. The semi-transparent bands represent confidence intervals obtained by bootstrap-resampling of the approximately 220 action potentials within a trial. The dynamic gain magnitude at the frequency of the sine component $|G(f_s)|$ can be calculated as the ratio of rate modulation $A_v$ to current modulation $A_s$. The values obtained from C in that way are overlaid as circles with confidence bands. Note the close agreement of the results obtained with the two different methods and also the difference in confidence intervals. The right panel shows the corresponding results for the 11 individual trials (thin semi-transparent lines) as well as the overall results of this one cell (thicker opaque lines) with their much narrower confidence bands. **(F)** The data from Fig 5 can be used for a critical test of the linearity of neuronal encoding under the conditions of our experiments. The last 25 ms before each aEPSC signal can be considered as approximately stationary. APs fired in these periods were used to calculate the dynamic gain curves for correlation times of 2 and 10 ms. Using these (complex) dynamic gain functions G(f), it should be possible to predict the transient increase in firing rate in response to the artificial EPSC inputs. However, this prediction assumes that the assumption of linear encoding is met. **(G)** The gain curves derived from the periods between the aEPSCs (opaque lines, see also Fig 5) are similar to the gain curves that resulted from the experiments in Fig 4, which are shown here semi-transparent as reference. **(H)** The predicted transient increase in the firing rate is shown. Because slower background fluctuations lead to a dynamic gain curve with a wider bandwidth, they are predicted to lead to a larger but briefer response. **(I)** The predicted curves from H are offset to the average firing rates and overlaid onto the measured rate increases. This comparison shows a quantitative agreement. Wide bandwidth dynamic gain (orange, 10 ms) leads to a more salient response to an artificial EPSC of 20 pA amplitude. Scale bars in A, and F are 20 ms and 100 pA. Data DOI for (C) and (E): https://doi.org/10.25625/VQUKFU.
(TIF)

**S2 Fig. A computational model in which randomly generated EPSCs with a reversal potential of 0 mV and IPSCs with a reversal potential of −70 mV combine at a holding potential of −50 mV to form fluctuating background noise.** In the model, the frequencies and amplitudes of the excitatory and inhibitory synaptic conductances are the same. **(A)** It can be seen that, as the decay time constant of the EPSCs increases, the time correlation of the noise is markedly slowed. **(B)** By contrast, similar changes in the decay of the inhibitory synapses have a much smaller effect on the

characteristics of the noise because the driving force for inhibition is quite small at −50 mV. **(C)** Similarly, the noise characteristics is also not very sensitive to the rise time of the EPSCs (but see the minor widening of the autocorrelation at its peak).
(TIF)

**S3 Fig. Temporal characteristics of excitatory and inhibitory synaptic inputs from local networks are revealed by voltage clamp recordings and glutamate uncaging.** Data DOI for (B),(D), and (F): https://doi.org/10.25625/VQUKFU.
(TIF)

## Author contributions

**Conceptualization:** Omer Revah, Fred Wolf, Michael J. Gutnick, Andreas Neef.

**Data curation:** Omer Revah, Andreas Neef.

**Investigation:** Omer Revah, Michael J. Gutnick, Andreas Neef.

**Methodology:** Omer Revah, Fred Wolf, Michael J. Gutnick, Andreas Neef.

**Resources:** Fred Wolf, Michael J. Gutnick, Andreas Neef.

**Software:** Omer Revah, Andreas Neef.

**Validation:** Omer Revah, Andreas Neef.

**Visualization:** Omer Revah, Michael J. Gutnick, Andreas Neef.

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
