## [Editor Report · Decision Letter 0]

18 Jul 2025

Dear Dr Neef,

Thank you for submitting your manuscript entitled "Cortical population coding critically depends on fine-tuning of cell physiology" for consideration as a Research Article by PLOS Biology.

Your manuscript has now been evaluated by the PLOS Biology editorial staff, and I am writing to let you know that we would like to send your submission out for external peer review.

Once your full submission is complete, your paper will undergo a series of checks in preparation for peer review. After your manuscript has passed the checks it will be sent out for review. To provide the metadata for your submission, please Login to Editorial Manager (https://www.editorialmanager.com/pbiology) within two working days, i.e. by Jul 20 2025 11:59PM.

Kind regards,

Taylor

Taylor Hart, PhD,

Associate Editor

PLOS Biology

thart@plos.org

---

## [Decision Letter · Decision Letter 1]

16 Sep 2025

Dear Dr Neef,

Thank you for your patience while your manuscript "Cortical population coding critically depends on fine-tuning of cell physiology" was peer-reviewed at PLOS Biology. It has now been evaluated by the PLOS Biology editors, an Academic Editor with relevant expertise, and by several independent reviewers.

In light of the reviews, which you will find at the end of this email, we would like to invite you to revise the work to thoroughly address the reviewers' reports.

As you will see, the reviewers wrote that the paper represents an important contribution to the field. However, they raised concerns about several aspects, including the computational approach, missing methodological details and rationale, imprecise statements, and several areas in need of further discussion and engagement with the existing literature. In your revision, you should carefully consider the reviewers' comments and respond to them thoroughly.

Given the extent of revision needed, we cannot make a decision about publication until we have seen the revised manuscript and your response to the reviewers' comments. Your revised manuscript is likely to be sent for further evaluation by all or a subset of the reviewers.

**IMPORTANT - SUBMITTING YOUR REVISION**

*Re-submission Checklist*

*Published Peer Review*

*PLOS Data Policy*

*Blot and Gel Data Policy*

Sincerely,

Taylor

Taylor Hart, PhD,

Associate Editor

PLOS Biology

thart@plos.org

REVIEWS:

Reviewer #1: This paper introduces an interesting and important question: how neural populations can faithfully encode inputs despite background noise, and how neurophysiology impacts encoding. However, the major problem with the paper is that the populations in question never seem to interact with one another, either in the experiments or, from what I can tell from the methods, in the computational model. Therefore, what seems to be studied is a collection of independent neurons whose activity is summed to simulate a population, where "correlated noise" inputs take the place of recurrent connections in the population.

The authors would need to incorporate additional neurons in the modeled population (with explicit recurrent connections) to make this argument/analysis more convincing. The experimental results are nice. The fact that L4 neurons have poorer high-frequency tracking than L5 neurons is interesting and compelling, as is the result that slower time constants in the background noise signal let a neuron more faithfully follow high-frequency inputs. But again, it's a stretch to describe it as a population response. In conclusion, while the results, including the model, are indeed interesting and important, they don't quite show what the authors suggest, so the paper would need to either include additional experiments and simulations, or be rewritten to less ambitiously describe the implications of the results.

General Comments

1. "In cortex, individual neurons fire no more than a few action potentials (APs) each second" is really only true in rodents. NHPs, for example, have healthier and more robust firing rates.

2. The introduction should spend a few more sentences explaining dynamic gain curves/analysis (introduced in lines 31-33 on page 1) to make sure that the reader understands the following statements (e.g. lines 38-42) at least superficially, without having to review the cited papers. This concept is easy to understand once one looks at the methods, but still useful to not have to flip back and forth between sections.

3. Text states "Its amplitude was chosen to evoke irregular AP firing at a basal rate ν0=3 Hz (Fig. 1B)" but Figure 1A caption says: "postsynaptic neurons firing asynchronously at about 5Hz"

4. "background noise" driven by a larger population of neurons than N, where N is the size of the subpopulation? (in reading the rest of the paper, I see that N is the total population and there are no other neurons, so Figure 1A should be corrected).

5. The model is supposed to capture the projections patterns in L4 cortical neurons, but it was unclear from the text how realistic it is that the N neurons have the same input AND the same output to the readout neuron? Wouldn't a more realistic scenario involve some variability in the connectivity between inputs and output? The question of "How salient this minimal input is for the readout neuron …" will also depend on this question. (In reading the methods, I understand that there is no output neuron, and you sum together all of the activities in time bins, etc. Again, Fig. 1A is misleading).

6. Figure 1C,D when talking about population size, does it refer exclusively to the postsynaptic population? How does that interact with the size of the non-targeted neurons?

7. It would be helpful if Figure 1 had a panel with more detailed example of connections between the dendrites, inputs, etc. in order to help the reader more easily understand the relationship between parametric changes and Pdetect

8. Why does "Enlarging the population size (N) … reduced the fluctuations in population rate"?

9. What is "vector strength" in Figure 2G? What test was used for significance? (Again, clear in methods, needs 1 sentence description in results).

10. In figure 4 C and E, the black bars read "10k trials." Should just be 10?

11. The aEPSC (fig 5) was injected into the same neurons every 50 ms (20 Hz) over an extended period of time. Could that cause any plasticity over the timescale of the experiment?

12. Figure 5 convincingly shows that inputs during background noise w/ longer correlations are more likely to elicit temporally precise responses in the recorded neurons. But the 10 ms tau condition also looks like the background noise is overall lower amplitude (in 5A). Maybe that is an illusion, but could you quantify that amplitude as well (in addition to the overall population activity, which you showed remains the same). Also, what is the relationship between the amplitude of the background noise and the likelihood that an AP will be elicited? From the data, it's hard to tell if the 2m tau condition just reduces the total number of AP elicited OR if it ends up jittering them across the population so that they "average out" across time?

13. For Figure 5, you state that: "In these experiments, since the noise is uncorrelated across sweeps, each sweep may be considered the equivalent of a different L4 neuron in response to the same aEPSC." but that's not quite right, because the response of the neurons in a population are really going to be impacted by circuit-level activity in response to the stimulus, which isn't fully captured by the background noise correlations you are inputing. For example, in the model you developed to study this question, does the activation of just one neuron propagate through the network? In a real network, one excited neuron might end up inhibiting others in its cohort, possibly even others that receive direct input from the presynaptic neurons. So while the experiment correctly simulates a large population of independently activated neurons, it likely fails to capture the interactions that would be experienced in a full network. I realize that restricting the analysis to the 2ms following the input reduces/obliterates the impact of the network on the single-neuron responses, but what it wouldn't reduce is the impact on a downstream decoder whose job it is to interpret the patterns of inputs from that population.

14. Text states that "Each neuron has two distinct sources of input: extrinsic projections and massive local, recurrent connectivity." but there is no mention of the recurrent connections in the "Theoretical model" section of the methods. Instead, it seems like the recurrent connections are not explicit in the model, but are simulated by time-varying noise inputs?

Reviewer #2: The authors of the ms study the response properties of excitatory pyramidal cells in layer IV and V of the somatosensory barrel cortex in mice in vitro. They demonstrate that the larger cells in layer V have a larger frequency bandwidth than the smaller principle neurons in layer IV, i.e. they respond reliably to a sinusoidal input current with a modulation of their firing rate even when the stimulation frequency is as high as several hundred Hz. These results hold true for a short-correlated background noise (in the in-vitro experiment represented by a noise stimulus generated on a computer) but are drastically changed when a longer correlated noise (with a correlation time of 10ms) is used, for which also the cells in layer IV display a high-frequency transmission. Moreover, the authors then argue that synaptic filter times are indeed longer in layer IV than in layer V, also the number of neurons and their speed of reaching the firing threshold is adapted that even weak inputs such as a single thalamo-cortical input spike, going to a subpopulation of N cells, can reliably be detected.

The results presented in this paper, in particular the experimental ones, are remarkable and an important contribution to a problem that has been predominantly studied by theoreticians. I am in favour of publishing a suitably revised version of the ms that takes into account the questions and recommendations listed below.

Major issues

1) The authors study the role of the correlation time of the background fluctuations, however, as the source of correlation they name only synaptic filtering. In the theoretical literature,

one additional potential source of correlation are the recurrent activity itself, that, specifically at larger synaptic amplitudes, can give raise to long correlations exceeding those introduced by synaptic filter times, see e.g. Ostojic Nature Neurosci 2014. The authors make a point that in their in vitro recordings correlation times match the synaptic decay times, but they should still discuss this second (more complicated) source of correlations and argue why - or, better, under which conditions - the synaptic filtering will dominate the correlation time.

2) On p.2 the authors write:

"In cortical networks, the postsynaptic effect of local, recurrent inputs reflects a balance of inhibition and excitation, and it is largely asynchronous from neuron to neuron."

This might be true but the weak remaining cross-correlations can have a strong impact on the population activity (see the famous contribution on this problem by the Bialek group, Schneidman et al. Nature 2006). So it would be important to state how strongly the L4 neurons are correlated exactly. And one could take into account how weakly correlated noise sources to different L4 neurons could mimic this effect. At least this problem should be mentioned and the limitations of the current approach (based solely on single-cell activity) should be pointed out.

3) Statements about the DC, AC, and noise injections should be completed with a table of the used values.

The authors write on p.15

"We adjusted the input's standard deviation, σI, based on the input resistance of the neuron, to assure that the voltage fluctuations were similar in all neurons. Along with σI, we also scaled the amplitude of the sinusoidal component Asin=0.37·σI. A constant, direct current (DC) 〈〉 was added and adjusted to maintain a target firing rate of ~5 Hz."

This is not reproducable, because we do not know the SD of voltage fluctuations. Different SDs could in combination with different DC currents result in the same 5 Hz firing rate. I suggest to give all the values in a table in the supplementary material: neuron number, correlation time and SD of OUP input current, amplitude of sinusoidal component, DC current, measured mean firing rate, measured maximal amplitude of firing rate modulation. In this way the reader (and in the first place, this reviewer) would get an idea about i) the variability of cells, ii) the susceptibility of the voltage variance with respect to the noise amplitude, iii) the (non)linearity of the signal transmission.

4) The authors compare the stimulation by a (comparatively) weak periodic signal and the estimation of the linear response function from the vector strength to the estimate of the response function from the full input-output cross-spectrum between the (non-weak) Gaussian noise stimulus and the output spike train. That these functions are the same (at least for an sufficiently weak cosine stimulus) is, according to my understanding, a consequence of the Furutsu-Novikov theorem that has been recently (re)derived in the context of spiking neurons by Lindner Phys. Rev. Lett. 2023. This connection should be mentioned.

5) Why the cell with larger dendrite has a braoder bandwidth of transmission has been discussed at length in the paper by Ostojic et al. J Neurosci (2015). The paper is already cited in the paper but the mechanism should be briefly explained because the difference in bandwidth between layer IV and V cells is such an important starting point of the paper. Additional references that might be useful and merit citation in this context are those by Gowers and Richardson Phys. Rev. Res. (2023) and Gowers, Timofeeva, and Richardson PLoS Comput. Biol. (2020) who studied analytically response functions in spatially extended neuron models.

Minor issues

1)

Bad line break on lines 11/12:

The effect of backg

round synaptic noise correlation time on pDetect and on dynamic gain.

2) p.3

The number of postsynaptic neurons is in the caption of fig.1 referred to as N but later in the paper, in the caption of fig.5, by n. Should it not be the same letter?

2) p.4

The authors write:

"... the background synaptic noise in a neuron reflects the relatively uncorrelated

synaptic bombardment by excitatory and inhibitory local inputs ..."

where it is not clear whether 'relatively uncorrelated' is meant with respect to time or with respect to presynaptic neurons; I suspect (and would agree) with the latter interpretation - the authors should here use an unambiguous phrasing.

3) P.6

"To this end, embedded a single, time-locked artificial EPSC (aEPSC) in the random noise ..."

->

"To this end, we embedded a single, time-locked artificial EPSC (aEPSC) in the random noise ..."

4) The ms ends on p.9 with two words, i.e. an unfinished sentence:

"Our findings"

5)p.15

APs were detected as crossings of zero membrane voltage.

->

APs were detected as upward crossings of zero membrane voltage.

(you do not want to count downward crossings as an AP time instant)

6)p.16

"Fourier transform of the autocorrelation"

This is well-known as the power spectrum!

Reviewer #3: [these comments have been copied from the attached document]

This manuscript by Revah et al. investigates how the temporal fidelity of cortical population coding arises

from specific cellular and synaptic properties. Using dynamic gain analysis, the frequency-dependent linear

response of a neuron or population to input fluctuations, the authors combine theoretical modeling with

electrophysiological experiments in rodent barrel cortex. They focus on Layer 4 (L4) spiny stellate cells as a

“stable, defined population” receiving known thalamic input. The study tests predictions that a population’s

encoding speed depends on cell number, cell size, and background noise correlation time Key findings are

that L4 neurons indeed exhibit a broad high-frequency bandwidth and high response fidelity, enabled by

their small size and unusually long synaptic noise correlation time. These factors appear “fine-tuned”

together with the number of L4 neurons such that even a single thalamocortical spike in the input can be

reliably detected in the L4 population output. Additionally, pharmacological manipulation of K_V7 (M-type)

potassium channels shows that M-current activity modulates the speed of action potential initiation, thereby

adjusting the dynamic gain. This suggests that the fast relay function of L4 can be gated by cholinergic

(muscarinic) brain states. Overall, the authors provide a mechanistic link between cellular physiology (e.g.

spike initiation kinetics, synaptic kinetics) and the sub-millisecond precision of population coding in cortex.

The questions addressed are highly significant for understanding cortical information processing. While it

has been known that cortical neurons can encode surprisingly high-frequency input changes, this work

pinpoints how specific biophysical parameters are matched to achieve near-optimal encoding in an actual

cortical microcircuit.

While the study is compelling, a few weaknesses or points of clarification should be addressed:

1) A potential concern is how well the slice experiment results translate to the intact, awake brain. In vivo,

cortical neurons receive continuous barrage from ongoing activity, and neuromodulatory tone differs across

brain states. The authors should clarify the assumptions made when extrapolating from their in vitro “stable

population” to in vivo function. For instance, if the slice was in an artificial down-state (with very low

spontaneous activity except what was injected), how might an up-state or aroused condition (with higher

baseline synaptic noise) affect the dynamic gain? They partially address this by discussing muscarinic

modulation, but it would strengthen the paper to explicitly note that in awake conditions L4 might already be

in a high-conductance state – and to speculate how that interacts with their findings. In particular, would the

“single thalamic spike → output” rule still hold amidst ongoing cortical activity? If possible, the authors could

discuss evidence from in vivo studies (e.g. Bruno & Sakmann 2006 showing reliable L4 spiking to single

thalamic events) to support that their slice results reflect real function.

2) The claim that even one thalamocortical spike is reflected in population output is exciting but could be

interpreted in different ways. Does it mean that one presynaptic spike in one VB thalamic neuron can trigger

at least one spike in L4 (across the population) on essentially every occurrence? This would imply an

extremely high sensitivity. Classical studies suggest thalamic inputs are weak individually, and a few

synchronous thalamic spikes are needed to drive an L4 cell to fire. The authors should clarify this point. It

may be that within a barrel, because there are ~100 spiny stellates receiving partially redundant thalamic

afferents, the population as a whole fires at least one spike for each thalamic spike (even if any given single

neuron is probabilistic). If that is the interpretation, the authors might explicitly state it to avoid confusion. As

a suggestion, providing a quantitative estimate of L4 population spike probability per thalamic input (and

how that compares to chance) would help substantiate the “reliable relay” claim. If this is based on modeling

rather than direct measurement, that should be stated. In any case, a clearer explanation would strengthen

the reader’s confidence that this is not an overstatement.

3) The finding that L4’s background noise has a longer correlation time than other layers is intriguing. The

manuscript would benefit from a more detailed discussion of why this is the case biophysically. Do L4 cells

have different receptor compositions or network connectivity that prolong synaptic events? For example, is it

due to a higher NMDA/AMPA ratio, or more GABA_B involvement, or simply that most “noise” in L4 comes

from recurrent activity with longer integrative times? Citing specific data (perhaps from their recordings of

synaptic kinetics, or known receptor differences) would support this point. In the text, they mention “synaptic

receptor dynamics” as the cause. It would help readers if the authors specify which receptors dominate L4

noise, e.g. “slow NMDA-mediated EPSPs and GABAergic currents lengthen the autocorrelation of L4

synaptic input.” This also ties into literature: slow glutamate receptors can sustain depolarizations and thus

increase the integration window. A brief comparison to layers 2/3 or 5 (which likely have faster

AMPA-dominated transients) could contextualize the uniqueness of L4.

4) The demonstration with K_V7 channels is convincing, but a few clarifications are warranted. K_V7

(M-type) channels are one of several subthreshold conductances (others include HCN/I_h and persistent

Na^+). The authors focus on K_V7 likely because their effect was strong, but it would be useful to

acknowledge whether other channels were considered or could play a role. For instance, I_h currents can

also influence responsiveness by depolarizing neurons and speeding up rebound firing. Did the authors

assume I_h was negligible in these L4 cells or kept constant? A sentence discussing this would show

thoroughness. Additionally, XE991 (the M-channel blocker) can have off-target effects at high

concentrations; the authors should note the concentration used and any steps taken to ensure changes in

gain were indeed due to M-current block (e.g. reversibility, known effects on input resistance, etc.). Since

the manuscript’s main claim here is that M-currents modulate spike initiation speed, they might consider

directly correlating the AP onset rapidness (slope of rise to threshold) before and after M-channel block.

This would link to prior work showing that AP onset dynamics govern high-frequency encoding. Providing

such a correlation (if available) would further strengthen the mechanistic point.

5) The authors repeatedly use terms like “optimized” and “fine-tuned.” While they do demonstrate that L4

meets certain theoretical predictions, it would be beneficial to see how sensitive the encoding performance

is to deviations in those parameters. For example, if L4 cell soma size were 20% larger, or if synaptic time

constants were 50% faster, does the high-frequency cutoff drop precipitously? The manuscript would benefit

from either data or discussion on this point – essentially, an analysis of robustness. Such information would

convey how precisely tuned the system is and could reveal which factor is most critical. It also helps readers

appreciate whether evolution had to precisely adjust all factors or if there’s some tolerance. If the authors

did not quantitatively do this, they should at least qualitatively discuss it, given it’s central to their claim of

fine-tuning.

6) In the Discussion, the authors should amplify the implications of their findings for sensory processing and

behavior. They touch on this (e.g. mentioning that coding in L4 may vary with brain state, but they could go

further. For instance, the fact that L4 can reliably transmit single spikes suggests the cortex can operate on

a single-spike per neuron basis for initial sensory detection. This resonates with theories of fast feedforward

processing in vision and touch, where very few spikes are needed for perception. The authors might

speculate on how the fine-tuning in L4 could be a general principle in other sensory cortices, or discuss if

similar optimization might be found in auditory or visual L4 (noting differences, e.g. auditory cortex lacks a

discrete L4 in rodents. Additionally, since they link to muscarinic control, they could mention potential

behavioral scenarios: during attentive whisking, cholinergic tone is high, which might transiently enhance

high-frequency transmission in barrel cortex, possibly improving detection of rapid whisker vibrations or

changes in texture. Making these connections would strengthen the significance of the work for a broad

neuroscience audience.

Reviewer #4: This is a very nice paper tying together the biophysical properties of individual neurons with a population's capacity to respond to inputs and transfer information.

My own expertise is in experimental measurement rather than computational analysis, and this is reflected in my questions/issues. They are all relatively minor.

1. What led the authors to choose 3 Hz as basal rate in their simulations (and 5 Hz in their experimental noise injections)? Was this picked as a physiologically realistic value for L4 spiny stellates? Please indicate if so.

2. In figures 2G and 4G, the statistical testing seems to rely on a battery of pairwise comparisons with no correction for repeated testing. I realize that some of the p-values in 4G are highly significant, but what is the rationale for not implementing a more robust approach correcting for multiple comparisons?

3. How did the glutamate uncaging experiments ensure that there was no direct "contact" with the cell under study? It is said that "special care was taken" but which steps ensured this are not clear.

4. How does dendrite length produce its effect on dynamic gain bandwidth? The theoretical papers looking at this issue suggested impedance load as a key factor, and intuitively one would expect input resistance to play a major role. And in the results from the experimental L5 vs L4 comparison results it is hard to disentangle dendrite length from Rin. Can the authors comment on this? There is essentially no reference to this in the paper.

5. What is the linear fit in Fig. 3D? From its appearance and judging by the caption, it seems to be a fit to the full data set (i.e. L4 and L5). I'm not sure if this makes sense - there is no obvious reason why all the points (for different cell populations with different biophysics) should be on the same line, and the constituent data sets are too small to test this properly. More superficially, if the fit is to the full data set, why is it plotted in the color of the L5 data?

6. Fig. 2D and E: are the grey lines actual best fits or just a visual guide added by hand? (and this also applies to Fig. 4C and D)

7. P. 8, lines 37-38: "further demonstrate that … EPSC decay time constants". I'm not sure the authors have fully demonstrated this. "Our findings indicate", perhaps?

8. P. 9, line 23: The discussion section seems to stop in mid-flow. Has part of it gone missing?

---

## [Decision Letter · Decision Letter 2]

3 Feb 2026

Dear Dr Neef,

Thank you for your patience while we considered your revised manuscript "Cortical population coding critically depends on fine-tuning of cell physiology" for consideration as a Research Article at PLOS Biology. Your revised study has now been evaluated by the PLOS Biology editors, the Academic Editor, and several of the original reviewers. We apologize again for the delay in sharing this decision with you, which was related to the need to discuss several points with the Academic Editor.

As you will see, the reviewers wrote that the manuscript is still in need of revision. Reviewer 2 outlined remaining concerns about the contributions of noise and cross correlations that may require additional analyses, as well as issues with the references. Reviewers 2 and 4 pointed out other areas requiring clarifications. Reviewer 1 commented that the title and abstract are overly broad. You should revise the manuscript to fully address the reviewers' remaining points, including through incorporating points from the rebuttal letter unless there is a compelling reason not to. This also applies to Reviewer 3's points that were not already addressed in the text.

We will then assess your revised manuscript and your response to the reviewers' comments with our Academic Editor aiming to avoid further rounds of peer-review, although we might need to consult with the reviewers, depending on the nature of the revisions.

**IMPORTANT - SUBMITTING YOUR REVISION**

*Resubmission Checklist*

*Published Peer Review*

*PLOS Data Policy*

*Blot and Gel Data Policy*

Sincerely,

Taylor

Taylor Hart, PhD,

Associate Editor

PLOS Biology

thart@plos.org

REVIEWS:

Reviewer #1: The authors thoroughly address all of my more minor general comments; however, the problem remains that the title and abstract of the paper do not accurately reflect their work. Regardless of the fact that the first 1-2 ms reflect primarily feed-forward inputs (a statement with which I have no argument), the authors ignore the significant connotations of the term "cortical population," which its related implications. Your title "Cortical population coding critically depends on fine-tuning of cell physiology" does not reflect that those implications are violated in this study. It well established that thalamic inputs constitute a minor fraction of cortical encoding of stimuli (Lien 2013), so considering subsequent cortical interactions is usually implicit in a paper titled "Cortical population coding….." If you are not addressing this point, the title and abstract should be modified to very clearly state the context you are considering: Early cortical responses to feed-forward thalamic inputs. Nowhere in the abstract is this limitation stated.

Reviewer #2: Please see the attached pdf file.

Reviewer #4: I am still convinced of the merits of this paper and support publication of a version with minor corrections.

1. The authors haven't quite addressed my question about their choice of a 3 Hz firing rate regime. They say the choice was not prompted by an established physiological value. It would be enough to say in the paper why they chose the value. Were they seeking a particular spiking regime? Additionally, could they sort out the apparent inconsistency between 3 Hz and 5 Hz in the text and figure caption, as also noted by another reviewer?

2. It would be useful if the clarifications on what "special care was taken" in the uncaging experiments were also described in the paper, not just in the response to my comment, for the benefit of readers.

---

## [Editor Report · Decision Letter 3]

18 Mar 2026

Dear Dr Neef,

Thank you for your patience while we considered your revised manuscript "Coding in cortical subpopulations critically depends on fine-tuning of cell physiology" for publication as a Research Article at PLOS Biology. This revised version of your manuscript has been evaluated by the PLOS Biology editors and the Academic Editor.

Based on our Academic Editor's assessment of your revision, we are likely to accept this manuscript for publication, provided you satisfactorily address the remaining points from the Academic Editor about the title and abstract (see at the end of this message). Please also make sure to address the following data and other policy-related requests.

IMPORTANT: Please ensure that your next revision addresses all of the following editorial points:

**Title:

As you'll see, the Academic Editor agreed with Reviewer 1 that the wording in the title should be more clearly defined in relation to the experimental context. We therefore propose the following alternative title options. Is either of these acceptable to you? As an additional note, if you propose something else for the title, we will need to discuss that with the Academic Editor, per their request.

Option 1:

Encoding performance of cortical neurons critically depends on their morphological and neurophysiological parameters

Option 2:

Encoding performance of cortical neurons critically depends on their morphological and neurophysiological properties

**Financial disclosure statement:

Please add links to the funding agencies in the Financial Disclosure statement in the manuscript details.

**Ethics:

Please move all information about ethical approvals to a new first subheading of the Materials and Methods section, entitled "Ethics Statement". Please include the specific national or international regulations/guidelines to which your animal care and use protocol adhered. Please note that institutional or accreditation organization guidelines (such as AAALAC) do not meet this requirement. Please also include the protocol/permit/project license number.

**Supplement:

Please ensure that you upload the latest version of the supplementary items (S1-S3 Figs; S1-S2 Table) as they were not included in the most recent revision.

**Data:

We require the numerical data underlying the figures. Please supply the numerical values either in a supplementary excel file titled "S1 Data" (filename: S1_Data.xlsx) or as a permanent DOI’d deposition for the following figure panels:

2G

3D

4GF

6B

S1C

S2

S3BDF

Please cite the location of the data clearly in all relevant main and supplementary Figure legends, e.g. “The data underlying this Figure can be found in S1 Data” or “The data underlying this Figure can be found in https://doi.org/10.5281/zenodo.XXXXX”

**Code:

Please note that our policy requires that all custom code or scripts be made available. Thank you for providing the underlying code in GitHub. However, because Github depositions can be readily changed or deleted, please make a permanent DOI’d copy (e.g. in Zenodo) and provide this URL in the manuscript and Data Availability Statement.

We expect to receive your revised manuscript within two weeks.

*Published Peer Review History*

*Press*

Sincerely,

Taylor

Taylor Hart, PhD,

Associate Editor

thart@plos.org

PLOS Biology

COMMENTS FROM THE ACADEMIC EDITOR [lightly edited]:

There is still a big problem with the title! I see the criticism that the authors record responses of a single cell in slice to repetitions of the same stimulus and draw conclusions about the transmission of this signal by a whole group of cells (a neural population) in the intact cortex in vivo. The authors argue that they are only interested in the early response for which (because of low spontaneous rates) lateral connections are unimportant and summing a thousand sweeps of stimulation is equivalent to the response of a (homogeneous) population of independent neurons. There are assumptions here about the independence and about the homogeneity of the system that are strong! I agree with R1 that using population coding in the title and in the abstract without any clear indication of these limitations is not advisable.

I do not see how the change to cortical subpopulations, suggested by the authors helps here.

[the following is a recommended modification of the abstract from the Academic Editor. The modified wording is surrounded by **]:

Sixty years after the concept of population coding in neuronal networks was introduced, we still lack a comprehensive understanding of its performance limits and the role of neuronal physiology. Here, we use dynamic gain analysis in a general model of population coding and demonstrate that disparate parameters of neurons and populations determine how accurately they can encode information. These are cell number, cell size, and the correlation time of the background noise. We experimentally test and confirm these predictions on **neurons of** excitatory populations in the mouse barrel cortex. Surprisingly, dendrite size and background correlations are precisely matched with the number of neurons in layer 4, such that even a single thalamocortical spike at the input is reliably reflected in the population output. However, this encoding performance can be modulated by the channels that mediate M-current, suggesting that coding in layer 4 may vary as a function of brain state.

---

## [Editor Report · Decision Letter 4]

14 Apr 2026

Dear Dr Neef,

As discussed separately, we invite you to submit another revised version of your manuscript after correcting some minor discrepancies related to Fig 2G. Please complete these changes and upload the updated files within one week.

*Published Peer Review History*

*Press*

Sincerely,

Taylor

Taylor Hart, PhD,

Associate Editor

thart@plos.org

PLOS Biology

---

## [Editor Report · Decision Letter 5]

21 Apr 2026

Dear Dr Neef,

Thank you for the submission of your revised Research Article "Encoding performance of cortical neurons critically depends on their morphological and neurophysiological properties" for publication in PLOS Biology. On behalf of my colleagues and the Academic Editor, Benjamin Lindner, I am pleased to say that we can in principle accept your manuscript for publication, provided you address any remaining formatting and reporting issues. These will be detailed in an email you should receive within 2-3 business days from our colleagues in the journal operations team; no action is required from you until then. Please note that we will not be able to formally accept your manuscript and schedule it for publication until you have completed any requested changes.

PRESS

Sincerely,

Taylor

Taylor Hart, PhD,

Associate Editor

PLOS Biology

thart@plos.org